# Mental models use common neural spatial structure for spatial and abstract content

Katherine L. Alfred 1*, Andrew C. Connolly[1], Joshua S. Cetron[2] & David J.M. Kraemer 3

Mental models provide a cognitive framework allowing for spatially organizing information while reasoning about the world. However, transitive reasoning studies often rely on perception of stimuli that contain visible spatial features, allowing the possibility that associated neural representations are specific to inherently spatial content. Here, we test the hypothesis that neural representations of mental models generated through transitive reasoning rely on a frontoparietal network irrespective of the spatial nature of the stimulus content. Content within three models ranges from expressly visuospatial to abstract. All mental models participants generated were based on inferred relationships never directly observed. Here, using multivariate representational similarity analysis, we show that patterns representative of mental models were revealed in both superior parietal lobule and anterior prefrontal cortex and converged across stimulus types. These results support the conclusion that, independent of content, transitive reasoning using mental models relies on neural mechanisms associated with spatial cognition.

[1] Department of Psychological and Brain Sciences, Dartmouth College, 6207 Moore Hall, Hanover, NH 03755, USA. [2] Department of Psychology, Harvard University, William James Hall, 33 Kirkland Street, Cambridge, MA 02138, USA. [3] Department of Education, Dartmouth College, Raven House, 5 Maynard Street, Hanover, NH 03755, USA. *email: Katherine.l.alfred.gr@dartmouth.edu

Mental models, cognitive architectures containing relational properties that reflect the structure of problem or situation, are a critical component of reasoning[1]. These representations can be built up through experience or they can be inferred from sparse information of a situation. Mental models need to represent a diverse body of information, but it is not parsimonious to have distinct models for every type of content. Instead, it is likely there is a fractionate system that includes a common framework utilized by all problem types, as well as recruitment of content-specific support systems[1–5].

Studies of the neural representation of relational reasoning have frequently tested the prediction that relational information is presented in a spatial way, through the creation of a mental map or model. The SPL and aPFC (often rostrolateral prefrontal cortex; RLPFC), have been found to underlie relational and transitive reasoning[6–8]. Transitive reasoning tasks typically use spatial content, making it difficult to draw conclusions about whether SPL activity reflects processing of spatial task content, or whether SPL activity is reflective of an inferred relational space representing conceptual distance between items, like a mental model[9]. It is possible that SPL involvement is due to the use of spatial content in tasks (e.g., height) and transitive reasoning tasks with abstract content would result in reduced SPL involvement. Alternatively, transitive reasoning could be a spatial process regardless of content, and individuals construct a spatial mental model to organize information.

It is important to note the difference between "content" versus "structure" in the context of transitive reasoning problems. "Content" in this article refers to the domain of the specific materials (e.g., "taller than"). A problem might not have spatial content if an individual is reasoning about how expensive one painting is compared to another or ordering a set of objects with an arbitrary ranking dimension. This use of spatial content is contrasted with the idea of the spatial "structure" of a mental model, i.e., generating a mental model with meaningful spatial structure, regardless of the items. For example, each of the types of content described above, whether spatial or non-spatial in content, could be organized spatially from "most" on the left to "least" on the right.

In addition to confounding spatial content and structure, past research on transitive reasoning tends to use tasks in which participants can directly perceive spatial stimuli while reasoning. Whereas this approach is useful for studying the neural basis of the reasoning process, it is sub-optimal for identifying neural patterns that represent inferred mental models unconfounded by perceptual information. Instead, neural patterns representing inferred mental models can be better examined using representational similarity analysis (RSA) to directly measure the informational content of the reasoning problem, even when the participant is no longer viewing stimuli demonstrating the transitive relationship[10].

Although past research has clearly implicated frontoparietal activity in spatial representations of mental models, these studies were designed to examine univariate patterns of brain activity and were only able to speak to relative activity levels in different regions, using inherently spatial tasks. In order to determine the structure of the activity in past work, Alfred, Kraemer and Connolly[10] used representational similarity analysis to compare the patterns of activity to the predicted mental model that would be created by a given reasoning problem. Activity in the SPL, IPS and the right aPFC correlated with the predicted pattern of activity based on the relative heights of "people" the participants were reasoning about. These results support the conclusion that a parieto-frontal network underlies the maintenance and use of mental models created through inferential reasoning, though this is partially confounded with the spatial task.

Though the spatial content and proposed spatial structure are confounded in previous studies, we hypothesize that there is a common spatial representation for mental models created through transitive reasoning. To test this hypothesis, in this study we expanded the content types used in the transitive reasoning problems to separate spatial content from the hypothesized spatial representation of mental models. We also aimed to use a task similar in design to our previous task[10], in that the relationships between stimuli should not be displayed at any point—they need to be deduced. As in the previous study, only adjacent pairs of stimuli were ever presented. Participants needed to use transitive reasoning to infer distal relationships needed to construct the full structure of the accurate mental model. Moreover, the information required to infer the correct order of stimuli in the linear space was not present during the task, but rather this information was learned previously and then queried for the purposes of drawing a conclusion during the task.

To accomplish these aims, we modified our previous paradigm to include a range of stimuli that vary in their spatial content. Participants constructed mental models with information from three content dimensions: an inherently spatial content domain: the height of a person (e.g., Dylan is taller than Kevin), a content domain easily mappable to estimated numerical magnitude: value of an unfamiliar abstract painting (e.g., a Distap is more expensive than a Tobir), and an abstract content dimension based on a nonsense descriptor ("vilchiness") randomly associated with non-meaningful line drawings (e.g., Hectis is more vilchy than Storog).

We hypothesized there is a shared frontoparietal network that supports the structure of mental models, as well as additional content-specific regions recruited based on the content of that problem. Results from an average RSA map across the three content-type z-maps and a conjunction analysis support this hypothesis. We found the only brain regions to reliably represent the structure of the mental models across content type were the right intraparietal sulcus and the left inferior frontal gyrus, indicating a frontoparietal network of brain regions support mental models across content types for transitive reasoning problems.

## Results

**Behavioral performance.** Participants completed the Hierarchy Recall task in both of two half-hour training sessions and the fMRI session for each content type (24–48 h gap between first and second training sessions, and 24–48 h gap between second training session and fMRI session). For each of the three content types (Height, Price, and Abstract), a Spearman correlation was calculated to determine the correspondence accuracy between each participant's generated ranked list and the correct ranked list of the items in that content type. Because the traditional format of transitive reasoning tasks takes the form of "A is more than B, B is more than C, Therefore, A is more than C", participants were only exposed to adjacent pairs during training. The transitive reasoning comes from inferring the relationship between A and C across their shared partner, B. This aspect of transitive inference can then be tested to determine if participants reasoned correctly by querying participants for both the full ordered list (hierarchy) as well as asking about how specific pairs of items relate to each other (forced choice pairs, below). Participants were not given any feedback during any training session about if their inferred ordering was correct. In the first session, participants were averaging a correlation of $rs(18) = 0.96$ with the actual order for Height, $rs(18) = 0.94$ with the actual order for Price, and $rs(18) = 0.79$ with the actual order for Abstract. At the end of the second training session, participants had an average correlation of $rs(18) = 0.99$ for Height, $rs(18) = 0.98$ for Price, and $rs(18) = 0.92$ for Abstract. By the end of the scanner session, participants had an

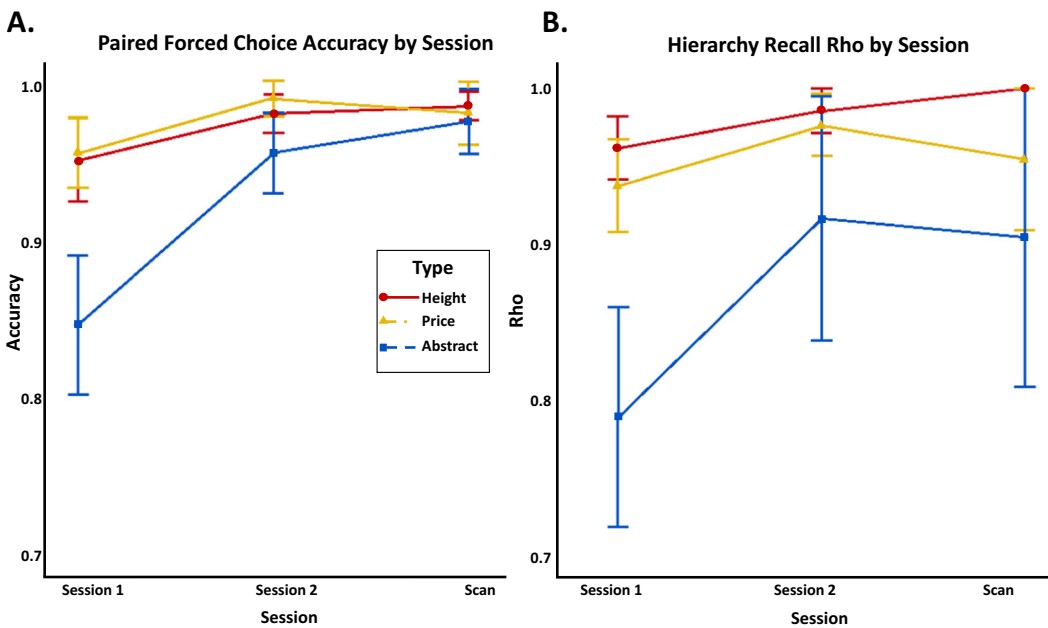

**Fig. 1 Behavioral results by task and session by content type.** All error bars display standard error (SE). **a** Accuracy for the Paired Forced Choice task by session. By the fMRI session, all participants had learned the hierarchies for each content type to criterion. **b** Spearman's rho for the hierarchy recall by session. Though the participant-generated rankings for the Abstract content type did not correlate with the correct rankings as well as Height or Price, participants had learned the spaces well enough that they were not consistently making errors.

average correlation of $rs(18) = 1$ for Height, $rs(18) = 0.96$ for Price, and $rs(18) = 0.91$ for Abstract (Fig. 1a). This result indicates that by the time that neural data was collected, participants had learned the hierarchy to ceiling. During the debriefing questionnaire after the fMRI session, participants were asked how they were thinking about the people, paintings, and objects and how each of those items related to all the other items in the space. The overwhelming majority (all except two) described using an explicitly spatial strategy, such as putting the items in the line, imagining them on a timeline, or creating a hierarchy. Of the two participants who did not use an explicitly spatial strategy, one of the participants reported numbering each of the items, and the other participant created a mental list using the first letter from each of the words. These results indicate that our transitive reasoning problems seem to inherently encourage the use of spatial structures to organize information.

Participants completed the Paired Forced Choice task in both training sessions and the fMRI session for each content type. Similar to the behavioral results from the Hierarchy Recall task, participants performed well on the Paired Forced Choice task. Participants completed this task during each of the three sessions for each of the three content types. Within each content type, participants were shown every possible combination of two items (including non-consecutively ranked items that were never presented together during learning). Participants were then asked to judge which of the two items was taller/more expensive/more vilchy. During the first session, participants had an average accuracy of 94.8% for Height, 95% for Price, and 85% for Abstract. In the second session, participants had an average accuracy of 97.6% for Height, 98.6% for Price, and 95.2% for Abstract. By the end of the scanner session, participants had an average accuracy of 98.1% for Height, 97.6% for Price, and 97.1% for Abstract (Fig. 1b; Supplementary Data 1).

**Neural representational similarity analysis (RSA).** A surface-based searchlight RSA[11] was conducted on the group level that reflected the Pearson correlation between local neural representational structure and a target similarity structure for each content condition (Fig. 2a). The RSA resulted in a correlation for each surface node for how closely the pattern of neural activity mirrored the pattern of the mental model, for each of the three content types. To correct for multiple comparisons, we conducted a 10,000-iteration permutation test. We z-scored the correlation with the a priori model at each node to this permuted distribution to find how likely it is that our results occurred by chance (Fig. 2b). The results of the permuted z-maps ($p < .01$, corrected) for each of the content types can be seen in Fig. 3. We further corrected the permuted z-maps at the cluster level to retain only spatially significant contiguous clusters ($p < .05$; see Method section for details).

For the Height content condition, we found $p < .01$ permutation-corrected probability for the RSA correlations in the predicted right IPS[3,10,12] as well as the right precuneus[10]. We additionally found a region of the right IFG[10], which may be involved in the retrieval of face-related information[13–15]. For the Price content condition, we found $p < .01$ permutation-corrected probability for the RSA correlations in the predicted right parietal lobule (superior and inferior parietal lobules)[3,10,12] as well as an inferior portion of the bilateral precuneus (as found in Alfred et al. [10]). For the Abstract content condition, we found $p < .01$ permutation-corrected probability for the RSA correlations as predicted in the right IPS[3,10,12] as well as the left IPS and the broader left SPL. Full results can be seen in Fig. 3. The left IPS has been found to play a similar role to the right IPS, though it typically is associated with symbolic mathematical processing as opposed to magnitude estimations[16,17]. It is possible that since the abstract content dimension was made up of novel abstract line drawings with nonword labels, there was no existing system for retrieval of these items (such as with the right IFG and face-retrieval), and participants were representing the items as symbols. It is important to note that while the content in the Height condition is more spatial than the Price condition, which is more spatial than the Abstract condition, we did not see any significant differences in the average Z-values in the right IPS between conditions, indicating that the right IPS had a similar

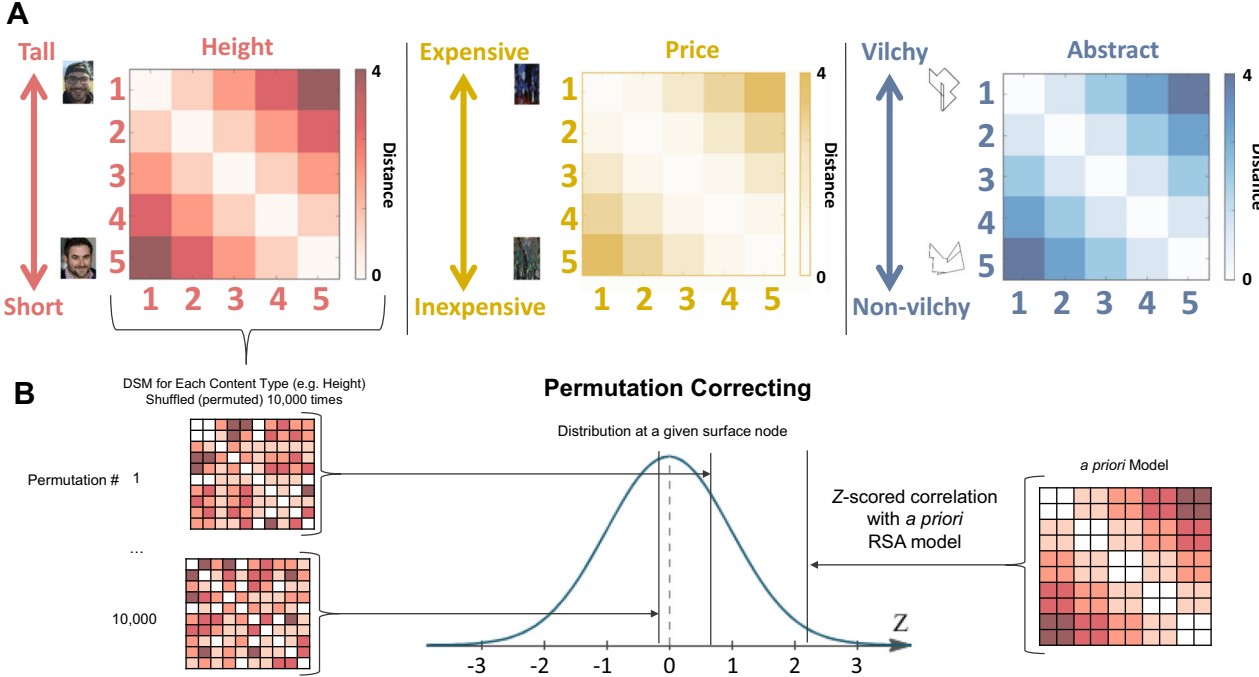

**Fig. 2 a** Dissimilarity matrices for each condition. Each of the items within each content type was modeled to be 1 rank distance from its neighboring objects. The diagonal for each of the matrices is 0 and was excluded from analyses. **b** Method of correcting for multiple comparisons. For each content type, the dissimilarity matrix was permuted 10,000 times to create a distribution based on possible outcomes for the data. For each surface node, the actual correlation value resulting from the a priori model was z-scored using that distribution of potential correlation outcomes. This z-scored correlation value represents both correlation strength and reliability.

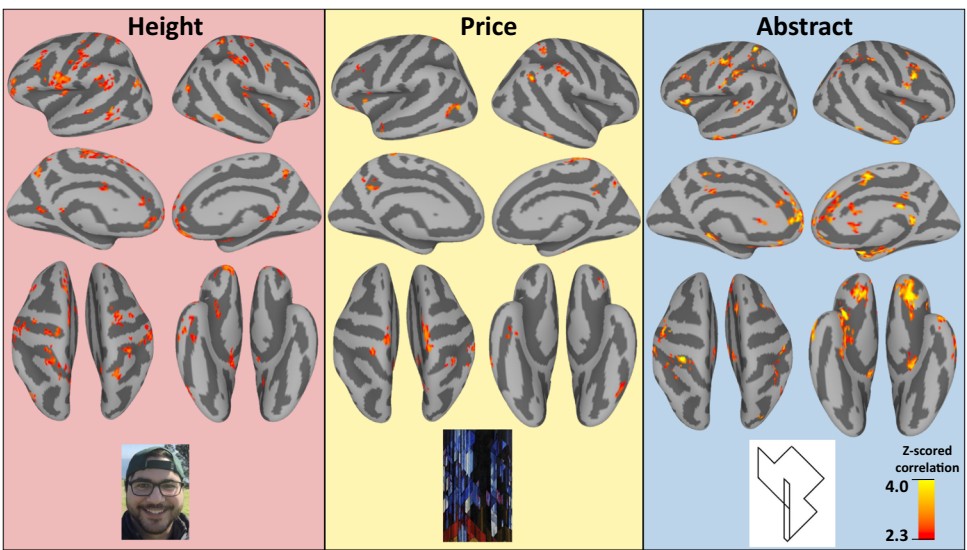

**Fig. 3 Permutation-corrected cortical surface maps for each of the representational similarity analyses by content type.** The z-scored correlations are indicative of correlation strength and reliability above the level of noise present in this dataset (permutation-corrected correlations, see Fig. 2B for further elaboration).

strength of representational similarity with the model in each of the conditions, regardless of problem content. This supports our hypothesis that the patterns of activity in the right IPS better represent information that is common between all the problem content types.

In short- the right IPS seems to be representing information about the common structure of the mental model constructed through transitive reasoning rather than the content of the specific transitive reasoning problems. All the problems used in

this study were transitive reasoning problems, which have a very specific organizational structure. Consistent with our hypotheses, there is a great deal of overlap in the neural localization of mental models across content types. Further, participants overwhelmingly reported using a spatial strategy to organize the information in the transitive reasoning problems, such as organizing the information on a line. Because nearly all participants used a spatial strategy for the transitive reasoning problems in all content types, we would predict to see similar

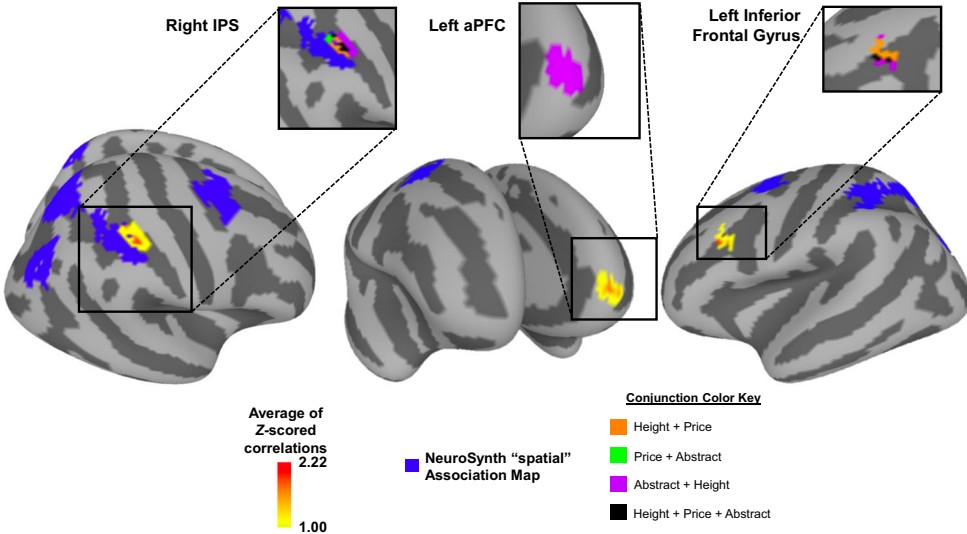

**Fig. 4 Common brain regions representing mental models across content types.** For all content-specific z-maps (Fig. 3), values below the significance threshold were removed. The three content-specific z-maps were then averaged (including 0 values for regions that did not reach significance in permutation correcting) to create the content-average map representing the common mental model structure across content types. Therefore, the z-values in this figure are averages of the permutation-corrected z-values from the content-specific maps (and as a result this scale is lower than that of Fig. 3). The average z-map was then cluster thresholded using a bootstrapped cluster significance level of $p < .05$, corrected (minimum area = 120 mm$^2$), so only values within significant clusters are displayed. The full cluster list can be seen in Table 1. Inserts display the conjunction map for each of the significant clusters. Both the right IPS cluster and the left inferior frontal cortex show overlap between all three content types. This average map was overlaid on top of term-based automated meta-analysis generated using NeuroSynth ("spatial" association map). The indicated right IPS cluster is the only cluster from the previous analysis that overlaps with the NeuroSynth "spatial" association map. See note in Table 1 for explanation of average z-value calculation.

**Table 1 Peak coordinates (MNI) and anatomical regions for average rank RSA.**

| Anatomical Region | X | Y | Z | Peak Z | Mean Z | Content Overlap |
|---|---|---|---|---|---|---|
| Intraparietal Sulcus (R) | 40 | −28 | 38 | 2.02 | 1.39 | Height, Price, Abstract |
| Inferior Frontal Cortex (L) | −40 | 22 | 28 | 2.06 | 1.43 | Height, Price, Abstract |
| Anterior Prefrontal Cortex (L) | −8 | 64 | 0 | 1.93 | 1.49 | Height, Abstract |

Note: Individual content RSA maps were thresholded at the node level based on our noise threshold calculations prior to averaging ($z > 1.65$). Cluster spatial extent thresholds were bootstrapped using AFNI's SurfClust function. Clusters have a minimum area of 120 mm$^2$, have a minimum distance of 3 mm between nodes in a cluster, and are significant, $p < .05$, corrected. The z-values reported here and in Fig. 4 are averages from all 3 content RSA maps which include zeros for any individual content map that has a sub-threshold value at that node

broad patterns of activity for all transitive reasoning problems unless the content itself was the primary determiner of patterns of neural activity.

To examine neural representations of mental model structure independent of content type, we identified regions present across all individual content RSA maps. After eliminating values that could have been boosted by chance through the averaging (see Method for details), we averaged the permutation corrected z-maps for all three content types. We identified significant clusters in the averaged z-map using AFNI's surface-based cluster simulation to determine minimum significant cluster area ($p < .05$; 120 mm$^2$). In the clusterized average z-map, we found a significant cluster in the predicted right IPS, as well as regions of prefrontal cortex (see Fig. 4 for whole brain map; see Table 1 for cluster list).

Because the average z-map reveals regions that significantly correlate with the content-specific rank models combined across content types, regions in this map are associated with mental models of a similar structure, rather than within a specific content type. It is important to note that despite the fact that the content varied from spatial to non-spatial, nearly all participants reported using spatial strategies to organize the structure of the information from these problems. Nonetheless, this analysis revealed a

significant cluster in the right IPS, in a region specifically associated with estimations of spatial magnitude in previous research[18,19]. To support our assertion that the involvement of the right IPS represents spatial information specifically, we used the association map from NeuroSynth with the "spatial" keyword (NeuroSynth.org[20]).

This map indicates areas that are selectively active for spatial information as compared to all other terms in the database (created through meta-analysis of 1157 studies that include "spatial" as compared to the remaining 13,214 studies, thresholded at FDR corrected .01). This approach is an objective, external, data-driven method to generate networks based on keywords in which activity across previous studies has been associated more with the term "spatial" than with other terms in the database. The spatial association map was binarized and the clusterized average z-map of the permutation-corrected and cluster-corrected RSA was overlaid on top of the spatial association map. The region in the right IPS is the only cluster that falls within the spatial association map, indicating that the right IPS is used for creating a spatial mental model of rank order created through transitive reasoning problems, regardless of content type used in the transitive reasoning problems (Fig. 4). Whereas a specific location in the rostrolateral prefrontal cortex

(RLPFC) is frequently reported in studies of transitive reasoning[6–8], we did not find that same region of the RLPFC to be significantly correlated with the average RSA for rank across content types. Instead, we found a cluster in the left inferior frontal cortex (IFC) that responded to mental model structure across content types. This region has been previously reported in reasoning research as a region that is active during correct analogy trials compared to baseline[8]. The cluster in the left IFC is also seen in our previous study using only the height-based transitive reasoning task[10]. We further identified a cluster in left anterior prefrontal cortex (aPFC) that is anterior and ventral to regions identified in the aforementioned studies demonstrating RLPFC involvement in transitive reasoning. Notably, these previous studies used tasks that differed from the current task in that participants were drawing new transitive inferences during the collection of neural data rather than querying a mental model previously created through transitive reasoning. In contrast, the left IFC and aPFC regions implicated by the current task relate to the process of querying a mental model previously created through transitive reasoning processes.

Finally, to determine which regions in the average RSA analysis represented similar mental models in each of the three content types, we calculated a conjunction map. This conjunction map highlights regions where either two or three content types had significant values after the initial permutation-based thresholding step. As with the average RSA, following a node-level thresholding step ($z > 1.65$), we applied a bootstrapped spatial cluster correction to the conjunction map to only preserve clusters significant at $p < .05$, corrected. Only the clusters in the right IPS and the left IFC were both significantly large and contained overlap between all three content types. The extent of the three-way overlap in right IPS is limited, however, the area surrounding that overlap is made up of different two-content overlap sections, indicating that adjacent and partially-overlapping regions of this region are coding these spatial mental models. Similarly, the cluster in the left IFC shows sparse three-way overlap, but it is surrounded by Height-Price and Height-Abstract overlapping regions, indicating the left IFC encodes the structure of the mental model across all three content types in the same region, if not precisely the same voxels within that region. The right IPS is the only region that overlaps with the NeuroSynth "spatial" meta-analytic map, consistent with the hypothesis that this region is involved in encoding the spatial structure of mental models across all three content types. The only other region to emerge in both the conjunction analysis and the average RSA analysis is the aPFC region that shows overlap between the Height and Abstract conditions, but not Price. No other regions survived thresholding for either the conjunction analysis or the average RSA analysis.

## Discussion

These results reveal that the structure of the mental model itself utilizes spatial machinery to encode and represent structural information about relational relationships, regardless of the content of the reasoning problem. Further, rather than relying on univariate contrasts, these findings are the result of directly modeling the patterns of neural activity using the representational similarity of the structure of the mental model itself, as it would be created through transitive reasoning, across both spatial and non-spatial content. Broadly, these results support the role of superior parietal cortex, and to a lesser extent anterior prefrontal cortex, in the creation and maintenance of mental models created through transitive reasoning across a variety of content types. An average $z$-map for rank (averaged across content condition) correlated with patterns of neural activity in the right IPS and anterior prefrontal cortex (aPFC). A conjunction analysis further

narrowed down these results to indicate that the right IPS showed significant levels of overlap between content types, including three-way overlap between all content types. Finally, the right IPS also overlaps with the NeuroSynth meta-analytic "spatial" map, indicating that the region of the right IPS that encodes all three content types has also been demonstrated by prior work to be a spatial region. This result indicates that the IPS supports the spatial representation of mental models created through transitive reasoning even when the content is not spatial. Not only is the mental model represented spatially when the content is spatial, but the same representation is created when the content is abstract.

These findings build on results from prior studies[12,21] that both concrete and abstract spatial information are represented the same way in the parietal cortex. The involvement of the aPFC supports previous work that shows prefrontal cortex involvement in the integration of relational reasoning[22,23]. Both the SPL and aPFC were implicated in the creation of mental models in transitive reasoning tasks using spatial content[6–8,10], and the involvement of those regions across a spectrum of spatial to non-spatial content indicates that the involvement of the IPS is due to the spatial structure of the representational space and not the spatial content of the problems. Further, within each content type, the models of each problem space correlated with additional regions in the superior parietal cortex, as well as content-specific support regions based on content type. This result is in line with a system of reasoning that both has shared processing for different problems, but also recruits different content-specific regions, depending on the specific problem content[2,3,23].

This study expands on a framework for relational reasoning put forth by Wendelken and colleagues[23] focusing on the role of the aPFC and SPL. Whereas this framework is broadly supported by the current body of research, there is some evidence the SPL may play a greater role than simply encoding individual relationships. The result from this study indicating that only the right IPS shows significant overlap between all three content types while the aPFC fails to reach significance indicates that the role of the SPL might be larger than originally posited. Specifically, the SPL may be critical for generating transitive inference itself. In one study of patients with aPFC or SPL lesions performing transitive reasoning tasks, only patients with SPL lesions showed significant impairment compared to aPFC lesioned patients and healthy controls[7]. Interpreted under the framework proposed by Wendelken and colleagues[23], lesions to either the SPL or aPFC should both have produced significant impairment to transitive reasoning. Because patients with aPFC lesions were not significantly impaired during transitive reasoning tasks, the SPL seems to be playing a critical role in both individual relationship encoding as well as generating inferences based on transitive relationships. More recent research using representational similarity analysis to probe the representation of mental models has also supported the involvement of the SPL while drawing transitive inferences[10]. Therefore, it seems plausible that the SPL is active in supporting transitive reasoning at the point of drawing inferences.

Most previous studies of transitive reasoning involve visuospatial stimuli, either through direct comparisons of magnitudes or using spatially grounded stimuli (such as "taller than" or "to the left of"). These methods pose a potential problem for interpretation of selectively spatial areas such as regions of the IPS. Are those spatial regions responding to the presence of spatial content in the problem, or to the accessing of a spatially organized mental model? Though our previous work[10] indicated that the patterns of neural activity in the right IPS highly correlates with a model that directly represents the mental model of the problem space of a transitive reasoning problem, those results were weakened by using the inherently spatial height content

domain. These findings are able to clarify that though transitive reasoning about content types with varying degrees of spatializeability results in information about the mental model being represented in different regions of the superior parietal cortex, a region in the right IPS showed patterns of activity that were consistent across transitive reasoning problems, regardless of content type. Further, the region of the right IPS that was found to have patterns of activity that corresponded with the structure of the problem space regardless of if the content was spatial was found to be contained within the NeuroSynth "spatial" association map. This pattern of activity indicates that the structure of the mental model for the transitive reasoning problems across content types is spatial even when the content is not spatial.

It is important to note that while this study has focused on the frontoparietal representations if mental models, significant prior work has focused on the role of hippocampal and entorhinal cortex[24–32]. Although the involvement of the hippocampus in the support of mental maps of relational information is a critical finding (and has often been demonstrated using abstract nonspatial relationships), there is evidence that the hippocampus is not the sole region responsible for this process. Some studies[9,23,31,33,34] have identified the involvement of SPL and RLPFC alongside the hippocampus with regards to the encoding of mental models created through transitive reasoning. Specifically, in Wendelken et al.[23] the authors noted that the hippocampus was specifically implicated in the process of drawing a new transitive inference across learned previously-associated pairs. Rather than study new transitive inference, in this study, we were primarily interested in testing the prediction of mental model theory that models are separately created and queried[1]. Although characterized as querying a mental model, this stage of the reasoning process does still involve reasoning during the reconstruction of the hierarchy from the model. Because this study was not collecting neural data at the time of the initial transitive inference, based on the finding from Wendelken et al.[23], we chose to focus on the role of the parietal and prefrontal cortex.

While it is meaningful to show the role of the superior parietal cortex in the creation and support of mental models created through transitive reasoning tasks, this finding has some limitations. Notably, transitive reasoning is a special kind of reasoning that differs both from other types of deductive reasoning (such as set inclusion problems), and also from inductive reasoning. A large body of work has shown that transitive reasoning is likely supported by mental models, which is not necessarily the case for all other types of reasoning. Because transitive reasoning is a form of reasoning that specifically compares relative magnitudes of items, it is especially likely that this form of reasoning utilizes the same mechanisms that allow individuals to compare directly perceived differences in magnitude. Parkinson and colleagues[12] identified and tested this link and found that literal spatial distance (near and far from an individual), temporal distance (near and far in time from the present moment), and social distance (close friend or stranger to an individual) shared common spatial magnitude processing in the right inferior parietal lobule.

This finding of shared processing for literal and abstract distance is in line with theories put forth by Dehaene and Cohen[35] that proposed that complex human cognition, such as the processes that support mathematics and reasoning, are supported by existing neural circuits that performed the most similar functions to the new complex cognitive functions. Because mental models tend to be represented spatially, including as mental maps of a given problem space, the spatial structure of these models seems to frequently be represented in neural regions that encode literal spatial maps or spatial distance, such as the hippocampal place maps[24–32] and the right parietal cortex, especially the right IPS[7,9,10,12,31,34].

Although there were a variety of initial content types in each of the transitive reasoning problems, representations of the content of those problems converged in a region that represented the common information between the content types—a spatial representation of relative distances[12,21]. This hypothesis is further supported by participant self-report during debriefing. Regardless of the type of content used in the transitive reasoning problems, participants stated that they used a spatial strategy to organize the information in the problems, i.e., nearly all utilized a spatial mental model.

## Methods

**Participants**. For this study, we recruited nineteen undergraduate and graduate student participants (15 female, $M_{age} = 19.81$) who were right-handed, fluent in English, and with normal or corrected to normal vision. No participants had a history of neurological or psychiatric disorders. Informed consent from each participant was obtained prior to the start of the experiment. Participants were compensated with their choice of cash or course credit for their participation, in accordance with the guidelines set forth by Dartmouth Committee for the Protection of Human Subjects (CPHS), and this study was conducted with approval from CPHS 23887.

**Transitive Reasoning Task**

*Learning*. Participants were trained on transitive reasoning problems featuring three types of content (Fig. 5): relative height of fictitious "people", relative price of fictitious paintings, and an abstract dimension of the relative "vilchiness" of abstract line drawings. Each "person" exemplar in the Height condition consisted of a picture--each a male face with a closed mouth neutral expression, (NimStim[36]) paired with a name taken from a normed list of the most popular two-syllable names from the 1990s (https://www.ssa.gov). Paintings were abstract paintings created by František Kupka paired with a fake name from the list of pronounceable nonwords with no English roots[37]. The objects in the abstract condition were abstract fully-enclosed black and white line drawings paired with a fake name from the same list of pronounceable nonwords[37]. Participants were presented with statements that took the form, "[Item A] is [more/less] [tall/expensive/vilchy] than [Item B]" (Fig. 5). The statements were counterbalanced so that participants were asked to either identify exemplars associated with more than or less than spatial relationships. For example, participants saw that "Distap is more expensive than Tobir" as well as "Matthew is less tall than William" (Fig. 5). Participants were told to use these statements to try to determine the relative height, price, or vilchiness of each item in the group. Because this was a transitive reasoning task, participants were only presented with adjacent pairs of items, and had to reason about the entire hierarchy through their knowledge of other relationships. In order to deduce that A is more expensive than C, the participant must reason that A is more expensive than B, and B is more expensive than C, so A is more expensive than C. Participants were shown each pairwise relationship in each direction (within each of the 3 content types, there were 4 possible direct connections between 5 exemplars, for a total of 8 possible statements) a total of 4 times per reasoning task training session.

*Hierarchy Probe*. For five seconds, participants saw a screen instructing them to think about where the following item fit into its larger hierarchy (Fig. 5). The lineup then disappeared and the name and picture of one of the items appeared for 5 sec while the participant thought about the relative height, price, or vilchiness of the currently viewed item. The name and picture then disappeared, and the participants saw the following set of three statements in random order, using height as an example, "Average" "More Tall" "Less Tall" with the numbers 1, 2, and 3 below the statements. Participants were instructed to press the number that corresponded with the statement that best matched the height of the item they had just seen. There were two runs per content type, and each item was presented twice per run (i.e., each item was presented 4 times total) for a total of 10 trials per run and 20 trials per content type.

*Forced Choice Pairs*. In this portion of the task, participants were instructed that they would see two of the items they learned about presented together (Fig. 5). Unlike during the transitive reasoning task described above, each possible pairwise comparison of exemplars within each content type appeared during this task. Participants pressed one button to indicate that the exemplar on the left was more tall/expensive/vilchy, and another button to indicate that exemplar on the right was more tall/expensive/vilchy. The presentation of particular exemplars on left or right side of screen was counterbalanced. Each content type had 5 total items resulting in 10 unique pairings per content type. The data for the forced choice pairs were collected from one run each per content type.

*Hierarchy Reconstruction*. Participants were given a sheet of papers with three sections, each with a blank 1–5 numbered list and instructed to write the names of each of the items for each content type in order from most to least (Fig. 5). To aid

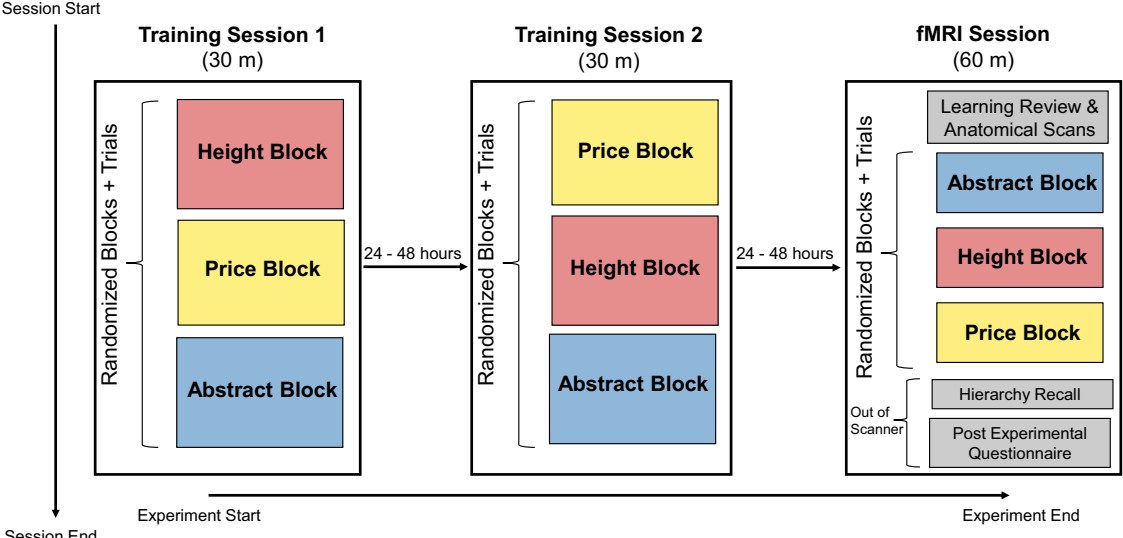

**Fig. 5 Overview of tasks and materials.** Each task (Learning, Hierarchy Probe, Forced Choice Pairs, and Hierarchy Reconstruction) was present in each of the three content conditions. The red area shows the version of the task with the Height condition, the green area shows the version for the Price condition, and the blue area shows the version for the Abstract condition.

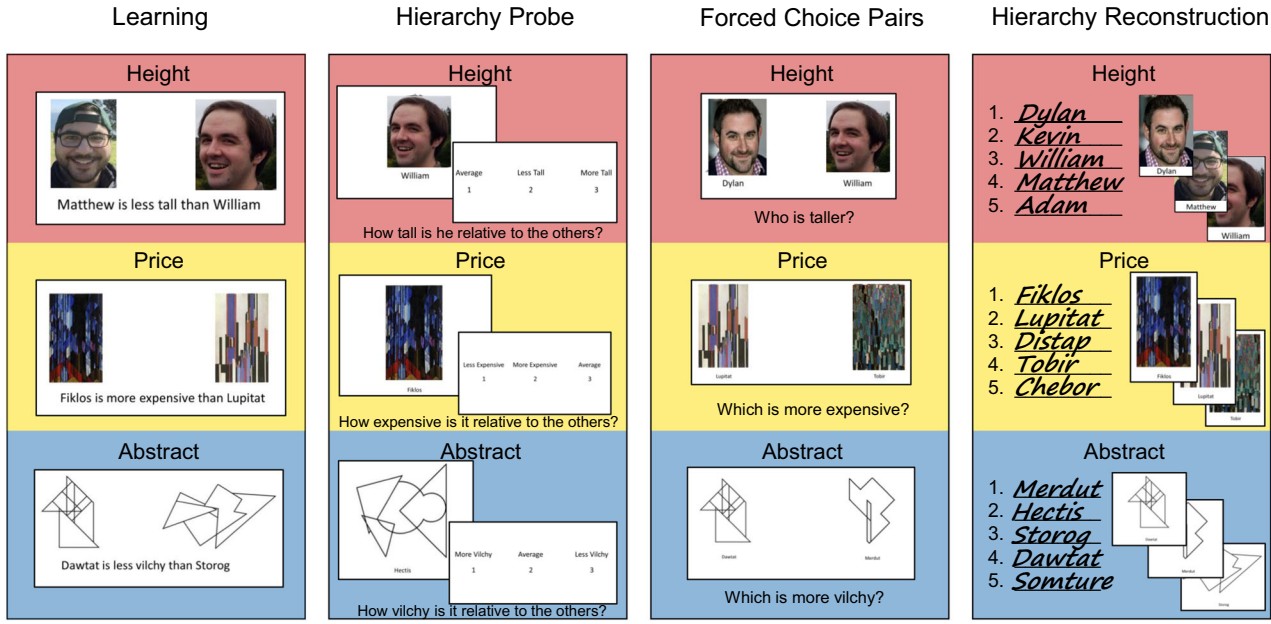

**Fig. 6 Experimental session timeline.** All participants underwent two half-hour training sessions where they were trained on the order of the items in each content domain. Within each content type block in each training session, participants completed a set of tasks in a fixed order to learn the hierarchy and practice the tasks they would need to do in the fMRI session. In each of the blocks in the Training Sessions, participants first observed pairwise comparisons of only adjacent pairs for each content type (e.g., comparing paintings 1 and 2, and not paintings 1 and 4). Participants then completed the Hierarchy Probe, Forced Choice Pairs, and the Hierarchy Recall before proceeding to the next content block. The ordering of these tasks within each content block in the training sessions was always in this set order. Both of the training sessions were 24–48 h apart and the second training session was 24–48 h before the fMRI session. In the fMRI session, participants completed a short review of the content presented in the Learning portion of the training sessions while anatomical scans were running at the beginning. In each content block, participants completed the Hierarchy Probe and Forced Choice Pairs tasks. Out of the scanner, participants completed the Hierarchy Recall for all content types and the Post Experimental Questionnaire.

participants that were largely relying on pictures and not names, an alphabetized (not rank ordered) guide of picture/name mappings was provided for participants to reference while organizing the exact ordering.

### Procedure
*Experiment Overview.* Figure 6 shows the overview of the experimental timeline, including both training sessions and the fMRI session. Each of the training sessions was a half hour in length. First, participants completed the transitive reasoning

task, and then completed the hierarchy probe, next they completed the pairwise comparisons, and lastly filled out the hierarchy on paper. The training sessions were required to be at least 24 h apart and no more than 48 h apart. The second training session had to be more than 24 h and less than 48 h before the scanning session.

*Training Sessions.* Each participant underwent two behavioral training sessions on two separate days to be familiarized with the task and to reason about the relative position of items in the hierarchies that they were reasoning about. First,

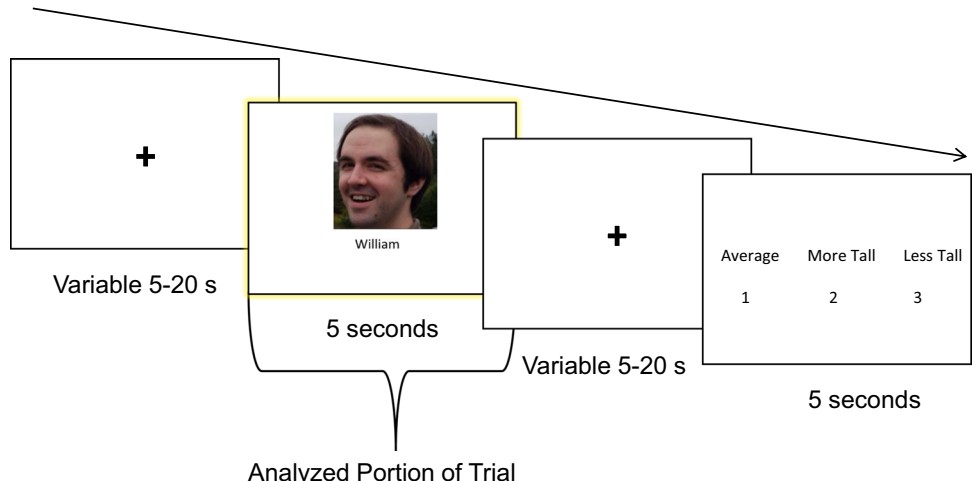

**Fig. 7 An example of one of the fMRI session hierarchy probe trials for the Height condition.** There was a variable length fixation period at the start of the trial (inter-trial interval), then the participant then saw the picture of one of the items they had reasoned about in training. There was another variable length fixation period, then participants were presented with a mapping of responses and button presses to respond. The mappings changed pseudo-randomly on each trial so that participants could not prepare a button response earlier in the trial. Only the 5 s period where the participant was being shown the item from the hierarchy and was considering its relative height was used in the imaging analysis. The fixation time was used as baseline for GLM comparisons of items to baseline. The button response portion of each trial was modeled as a regressor of no interest and not analyzed further.

participants were trained on adjacent items in the hierarchy for each of the height, price, and vilchy content domains. With price for example, participants were shown that one painting is more expensive or less expensive than another painting, and were reminded that "more expensive than" means "immediately more expensive than" and not just more expensive broadly than that other painting to indicate that there is a specific price order that can be deduced. Participants were instructed to pay close attention to how expensive each painting is relative to the rest of the group. Over the course of the task, each item was presented with both the exemplar more expensive and less expensive than it, and each comparison between two paintings was repeated four times over the session. After the initial learning session, participants were given a test where they were told to consider the placement of the following painting in the overall hierarchy. Then, one of the items was shown alone on the screen as the participant thought about the price of that item. The item was then removed from the screen, and the participant pressed a button to indicate whether that painting was Average, More Expensive, or Less Expensive compared to the group. The mappings between buttons and the statements did not appear until after the presentation of the item being considered in order to separate conclusions about the rank of that item from preparing a motor response or perception of spatial information. Participants then completed the Forced Choice Pairs task, where they were presented with two items, and asked to press F if the painting on the left is more expensive or J if the painting on the right is more expensive. This task compared every possible combination of items within each content type, including items that were never learned together, and whose relative prices could only have been deduced through transitive reasoning. Lastly, participants were given a blank sheet of paper divided into three sections (one for each of height, price, and vilchiness). Each list was numbered 1 through 5 and were asked to write the names of the items in order from most to least ("Hierarchy Reconstruction" task) for each of the three content types. Participants were provided with an item reference sheet that contained an alphabetized list of the items (which did not correspond with rank order in any way).

*MRI Scan Session Overview.* The MRI scanning session lasted for 1 h including the structural scans and had the same basic structure as training sessions. Participants reviewed the transitive reasoning task during anatomical scans at the beginning of the session. The hierarchy probe and pairwise comparison tasks were completed during functional runs (Fig. 7), with the two runs for the hierarchy probe (for each content type) occurring before the forced choice pairs task. During the scan session, each content type's hierarchy probe task was completed twice to maximize data points of participants thinking about the relative rank of each item. The two hierarchy probe and forced choice pair task were completed blocked by content type to prevent unnecessary memory load from switching item types. The ordering of the content blocks were counterbalanced between participants. The full hierarchy was completed again on paper immediately after the completion of the scanner study.

### Experimental Design and Statistical Analysis

*MRI Scanning parameters.* The MRI session took place at the Dartmouth Brain Imaging Center using a Phillips 3 T Achieva Intera with a 32 channel SENSE head

coil. For the functional runs, there were six (2[run]x[3 content type]) runs of 148 volumes per run for a total of 888 functional (T2*) volumes with a TR of 2.5 s. The functional scans were collected using gradient-echo EPI with 42 Philips interleaved transverse slices at 3 mm per slice (TE = 35, flip angle = 90 degrees).

*Univariate Functional Imaging Analysis.* This univariate analysis for the neural data from the Hierarchy Probe task (see Fig. 7 for example using Height condition) was conducted to obtain beta values for each item, used in the representational similarity analysis (discussed below). Each run of neural data was preprocessed separately with FSL tools for motion correction and registration (MCFLIRT[38]). Neural data sets for each run of each participant were modeled using a canonical HRF was used (6 s to peak) and were smoothed using a 5 mm FWHM Gaussian kernel. Neural responses for each of the exemplars (two repeats per exemplar) were modeled in a contrast against fixation baseline. The button response periods after the stimulus presentation window were also modeled and used as a regressor of non-interest. Each run for each participant was modeled separately. Because each participant completed the Hierarchy Probe session twice per content type, the GLM was initially analyzed using FSL FEAT on the level of the individual runs, and a higher level GLM was conducted to obtain item-level betas across runs within the same content type (e.g., to obtain the beta value for "Matthew", a higher level GLM was conducted to find the beta for "Matthew" across both Height Hierarchy Probe runs). Beta values were obtained through the contrast between each exemplar item and unmodeled baseline across both runs. In order to isolate neural data related to the task from neural data related to responding, participants were not told the mapping of buttons to responses at the start of the trial so motor responses could not be prepared until after the analyzed portion of the trial. Regressor covariance estimates generated by FSL (version 5.0) confirmed that these portions of the trial were statistically separable due to the jittered fixation periods inserted in between sections of each trial. Anatomical data for the searchlight portion of the analysis were prepared using FreeSurfer[39].

*Representational similarity analysis.* The following analyses were performed using Python and PyMVPA[40], SciPy, and NumPy. The searchlight-based representational similarity analysis (RSA) was conducted on the neural surface, using a 100 voxel searchlight mapping technique[11] that produced a whole-brain map on the group level (created using the average of the individual subjects untargeted neural similarity for each content condition) that reflected the Pearson correlation between local neural representational structure and a target similarity structure for each of the content conditions (Height, Price, and Abstract; Fig. 2). Each modeled dissimilarity matrix was created using the ordinal ranking of the objects, where the tallest/most expensive/"vilchiest" object is one distance away from the second object, two from the third object, and so on for all five items in each hierarchy. At each searchlight location, the local neural dissimilarity matrix was computed using correlation distance between activity patterns (derived from beta values in the item-level GLM, described above) for all pairs of stimuli within that content type. Activity patterns were defined by the voxel-wise estimated hemodynamic responses from GLM analysis of the functional data collected during the two Hierarchy Probe runs for each of the three content conditions (excluding the portion of the trial in which a button response occurred). After all individual untargeted neural

dissimilarity matrices were calculated, a group average was calculated within each content type. Each of these content-specific average neural maps were then correlated with the modeled dissimilarity matrices for item rankings within each content type. The resulting correlations were permutation-corrected against a null distribution of 10,000 randomized potential correlation maps, resulting in a corrected Z-map for each content type (Fig. 3) indicating the likelihood that any observed correlation was significantly different from chance. For further details, please see the section on multiple comparison correction below.

These three content-specific maps were further used as input for the higher-level Rank Average RSA analysis. In this analysis, we aimed to identify neural regions that showed patterns of activity that matched the predicted structure for the mental models across multiple content types. Through our generated distributions from our permutation corrections (described in detail below), we identified that the noise threshold was $z > 1.37$. Given we were testing our hypothesis about a positive correlation between the theoretical mental model and the pattern of neural activity, we set our cutoff for the threshold for minimum values from each of the permutation-corrected content-specific maps as $z > 1.65$, a one-tailed $p < 0.05$. In short, when the average rank $z$-map was calculated, only permutation-corrected $z$ values greater than 1.65 from each of the content-specific RSAs were included. All other values were set to 0 to prevent averages being artificially. The content-specific permutation-corrected $z$-maps that were thresholded at $z > 1.65$ were averaged together to create one average map that represented relative rank of items, across content types. We then used AFNI's surface-based cluster simulation to identify significant clusters on the surface (clusters would be required to have an area greater than $120 \, mm^2$, at a bootstrap-corrected threshold of $p < 0.05$, corrected).

After clusters were identified, we further verified that the clusters were not due to an exceptionally high value from a single content specific map. To do so, we identified that the maximum permutation-corrected $z$ that was included in the average was 4.1 Given the average map was calculated from three maps, an average between 4.1 and two additional 0 s (non-significant values from the other two maps were masked as 0) resulted in a $z$ of 1.36. This maximum value is the highest possible value that could have resulted from a single map contributing to a region in the average map. The cluster in the right IPS, left IFC, and the left aPFC both had peak and cluster-average $z$ values higher than that cutoff, indicating that both of those clusters could not have possibly been present due to the results from a single content-specific map, and minimally indicate that multiple types of content show the same pattern of activity in those regions.

Finally, to investigate which regions in the average RSA analysis represented similar mental models in each of the three content types, we calculated a conjunction map. Each of the original content-specific RSA maps were thresholded $z > 1.65$, $p < .05$ (as with the average map above). Then, each content-specific map was binarized, so that each node greater than the minimum threshold had a value of 1 and all other nodes were 0. The three binarized maps were then multiplied by different values and summed together. Different values were assigned to each map so it would be clear which content types were overlapping in which regions (which would be lost through simply binarizing the maps and summing).

*Multiple comparison correction.* We conducted a permutation test to compare our predicted dissimilarity matrices and observed results to a distribution of possible results based on a distribution of 10,000 random permutations of the target labels. The probabilities associated with our results were thus calculated as the $z$-scored likelihood of the actual $r$-values occurring by chance at a given node compared to a distribution of possible outcomes created by shuffling the data 10,000 times. In order to examine the effects of rank across content types, we averaged the permutation-corrected $z$-maps for each content type, thresholded at $z > 1.65$, one-tailed $p < 0.05$. This average map for rank was further bootstrap cluster corrected using AFNI's 3dClustSim function for surface clusters significant at $p < 0.05$ ($120 \, mm^2$).

*Statistics and reproducibility.* The searchlight representational similarity analysis $z$-maps for each content type (described in detail in the Representational Similarity Analysis subsection above) were permutation-corrected for multiple comparisons using distributions created from 10,000 random permutations of the actual data (see Multiple Comparison Correction section above). From this permuted distribution of data, we identified that our noise threshold was $z > 1.37$. We selected a more conservative threshold of $z > 1.65$ for inputs to our $z$-map averaged across all content conditions, and we further corrected that average $z$-map with ANFI's surface-based cluster correction to identify clusters with an extent significant at $p < 0.05$ ($120 \, mm^2$). The same correction parameters were applied to the conjunction analysis as well.

One condition of this study, the Height content condition, was included to directly reproduce our prior results reported in Alfred et al. [10]. We were able to successfully replicate our prior results, indicating that the neural signal associated with representation of a spatial mental model created through transitive reasoning is reliably reflected in patterns of neural activity in the right IPS.

**Reporting summary**. Further information on research design is available in the Nature Research Reporting Summary linked to this article.

## Data availability

Data will be made available upon request. Neural data are in the form of NIML surface datasets. Behavioral data are in comma separated value files (Supplementary Data 1). Requests can be made to the corresponding author.

## Code availability

Custom code for the permutation corrections will be made available upon request. Requests can be made to the corresponding author. All other analyses were conducted using tools provided by FSL, FreeSurfer, PyMVPA, Numpy, and Scipy, as outlined in the method section.

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

## Acknowledgements

Support was provided by a National Science Foundation award (DRL-1661088) to D.J.M.K.

## Author contributions

K.L.A. and D.J.M.K. designed the study, developed stimuli, collected data, performed analysis, and wrote the manuscript. J.S.C. additionally performed data cleaning and preprocessing, analysis and contributed significant python coding support. J.S.C. and A.C.C. contributed to study design, interpreting results, and helped with development of the analysis pipeline and the custom permutation correction code.

## Competing interests

The authors declare no competing interests.
