## [Peer Review File · Communications Biology]

Reviewers' comments:

Reviewer #1 (Remarks to the Author):

This multivariate fMRI study tests the hypothesis that the mental models that we construct in order to represent hierarchies among individual items, which we can then draw on to perform transitive inference, are inherently spatial. It builds nicely on a prior study in which the authors showed - using representational similarity analysis, or RSA - that right inferior parietal sulcus (IPS), a region strongly implicated in spatial representations, shows a pattern of activation consistent with the representation of an ordered hierarchy of stimuli. In that initial study, the learned comparisons among stimuli (the relative height of two individuals) were inherently spatial. The present study expands on that work by teaching participants two non-spatial hierarchies in addition to the spatial one. Once again, right IPS appeared as a key focus in the RSA analysis. Other clusters were also observed, including one in frontopolar cortex, but only the right IPS showed overlap with the Neurosynth meta-analysis for the term 'spatial'. This is a fabulous study. I have only two comments:

1) The authors note that some comparisons are more readily 'spatializeable' than others. As such, it would be nice to see whether right IPS shows a gradient in strength of representational similarity for height > time > the abstract quality ('vilchiness').

2) The final analysis reveals a cluster in frontopolar cortex; it is quite ventral and anterior to the region in anterior prefrontal cortex that has been implicated in univariate fMRI studies of transitive inference and other reasoning tasks (often referred to as rostralateral prefrontal cortex). This is worth noting explicitly. Perhaps rostralateral PFC is involved in the process of actively drawing comparisons on the fly, but not in drawing on representations from a learned hierarchy.

Reviewer #2 (Remarks to the Author):

The manuscript reports an experiment investigating the neural basis of transitive reasoning, when the type of ordinal relationship, and accompanying visual cues, are varied. The main finding is that across three "content" types transitive reasoning recruited some of the same brain regions, specifically IPS, which is seemingly involved in spatial reasoning. The authors conclude that, even when the ordinal relationship that is being reasoned about is not spatial, brain regions involved in spatial reasoning are still recruited.

Overall, I think this is an interesting, and well-designed study (e.g. the statistical analyses, in particular the permutation test, are appropriate given the experimental design), and the main finding is compelling. However I do have some concerns primarily about how the experimental design is characterized and the results are interpreted.

Main comments:

A. The authors contrast the height task with the price and abstract task as being the only one that involves "spatial" relationships. Indeed, a prime motivation for the study was to address a limitation of Alfred et al. (2018); namely, that it used an "inherently spatial content dimension" (L143). However, the height task does not involve the subjects ever actually seeing the heights of the different individuals. Instead, subjects make transitive inferences based solely on information about the ordinal relationship of adjacent individual heights in the hierarchy. Given this design, it is not totally clear to me why one would think, *prima facie*, that the spatial nature of the content would matter in the first place to what brain regions are recruited for carrying out transitive reasoning.

Coming from the other direction, it is quite natural to map ordinal relationships onto a spatial dimension (in our "mental model"). So while a price is not something that we can "see", a ranking of

prices is something that we might naturally reason about by imagining the artworks being put in a line based on increasing price. The same is plausibly true for the abstract condition, where we might treat shapes as symbols for representing different positions in a rank-order (which the authors at one point allude to).

The point I am trying to get to is this: it seems that ordinal relationships are ones that we naturally think of in spatial terms, and in that respect, all three tasks involve reasoning about relations that are potentially equally "spatial" in nature. In which case, it would make sense that they all recruit the same spatial brain areas, because transitive reasoning is a kind of relationship that maps nicely onto spatial relationships. The authors acknowledge this in the discussion when they say that "the involvement of [SPL and aPFC] across a spectrum of spatial and non-spatial content indicates that the involvement of the IPS is due to the spatial structure of the representational space and not the spatial content of the problems" (L323-325). But I think it would make sense to acknowledge this idea from the outset. More specifically, the authors could articulate explicitly two alternatives: (1) that if what matters is the spatial content, then we would expect to see differential activity in the price and abstract tasks (with the latter recruiting no spatial regions); and (2) if it is the spatial structure, we would expect to see similar spatial processing brain areas light up across tasks. I think being clearer about these two alternatives might help to clarify the novelty of the study.

B. The talk of "content" is a bit obscure. What is clearly meant is the "representational content", or what it is that is being represented when subjects are doing the transitive reasoning. However, per my previous comment, the content that subjects are really reasoning about is the ordinal structure, and that is identical across tasks. So I think it would be useful to disambiguate that there are two kinds of contents here: the subject matter, and the ordinal structure. It might also help readers (who are less used to seeing 'content' being used in a technical sense) to have a sentence defining what is meant when referring to task "content". Related to this is what is meant by "mental model" and whether it is spatial.

C. It seems one could draw a connection between these results and the issue of embodiment and the format of the representations that underlie transitive reasoning. If space permits in the discussion, it might be useful if the authors said a little bit about what implications these results have for that debate/topic (e.g. in relation to the final paragraph of the discussion, where transitive reasoning is contrasted with other kinds of reasoning, which might not require mental models).

D. I think a few more methodological details (see minor comments below about the searchlight, training task) should be mentioned in the results. I didn't quite understand the findings of the experiment until I had read through the methods as well, and then went back and read the results again.

Minor Comments

1. L51, unclear what is meant by "basic information".
2. L72-79, the reference to the "approach" here is strangely opaque. It should be described here explicitly.
3. L73, the embedded clause "which leverages...analysis" should probably have commas at the start and finish.
4. L110, what does "the region" refer to? SPL or IPS? Please disambiguate. I assume the authors mean IPS.
5. L121, sentence is unclear. What does "it" refer to in "it is associated"? SPL not V2 I assume?

6. L142-145, here is a case where one could simply refer to spatial "subject matter" rather than "content", if one was inclined to drop the content-task.
7. L208, from context one can determine the authors carried out a surface-based RSA searchlight, however as far as I can tell the authors do not explicitly state they are carrying out a searchlight analysis until the methods. Since results are coming before methods, a bit more details are needed here to describe exactly what the analysis is.
8. I would maybe consider combining Figure 5 with 4, and just show the NeuroSynth patches on the latter, since Figure 5 is just repeating a portion of 5, as far as I can tell.
9. L357, typo. Should be either "differs" or "is different".
10. L372, is this a typo? Should "materials and methods" be deleted?
11. L390, I think this is the first place it is mentioned that subjects only saw adjacent pairs during training. If so, this is an important methodological detail that should be mentioned in the results.
12. Why only have 3 options in the hierarchy probe when there are 5 items? Was this because of scanner limitations for performing the task in the scanner?
13. Figure 7, there is a lot of redundant text in this figure when depicting the training sessions. Perhaps the authors could try and simplify things. I would also include a vertical arrow to indicate that boxes at each stage are ordered by time vertically (e.g. for the fMRI session).
14. L526-541, some queries about RSA. Did the authors construct neural similarity or dissimilarity matrices? In the text it is a bit ambiguous. And why did the authors calculate the Pearson's r coefficient between behavioral and neural RDMS, since the rank-order correlation (Spearman's ρ or Kendall's τ) is used, since they do not assume linearity, only monotonicity (see. e.g. Kriegeskorte, Mur, and Bandettini, 2008)?

Reviewer #3 (Remarks to the Author):

In this manuscript, Alfred et al present an fMRI study examining the neural representation of mental models involved in relational reasoning across different content types. Using representational similarity analysis (RSA), they show overlapping representations of the transitive structure across content types in superior parietal lobule (SPL) and anterior prefrontal cortex (aPFC). In addition, SPL falls within clusters in the association map for the keyword "spatial" in NeuroSynth. They conclude that this reflects shared neural mechanisms with spatial processing.

This study asks an interesting question that echoes a growing trend in neuroscience. There has been an increasing attention on how spatial models could be utilized to represent abstract mental structures that are not intrinsically spatial, and whether they share similar neural substrates. The inclusion of three different content types is a nice feature of the design, and RSA also seems a suitable approach for the research question. However, despite such strengths, I believe that there still exist several significant expositional, methodological, and interpretive issues that need to be addressed. In particular, what contributions this study brings are a bit unclear to me, and it is also hard to evaluate how much the current results support the main conclusions because of the lack of clarity in the results and certain controls for alternative hypotheses. My detailed comments are as follows.

1. "Mental model" is a very broad term, which could mean a lot of different things in different fields. Given the rather specific type of mental model that the task requires, it is necessary for the authors to tone down the claims made in the title and throughout the article (esp. Introduction and Discussion) to something more precise such as "transitive structure" or "relational inference". More work also needs to be done to improve the writing of the Introduction and better position this study in the context of extant literature. Specifically, considering the broad audience of this journal, it will be helpful for the authors to mention existing studies showing spatial coding of fairly complicated non-spatial mental models already (e.g. Constantinescu et al., *Science*, 2016; Garvert et al., *eLife*, 2017; Schafer & Schiller, 2017, *Neuron*; Theves et al., 2019, *Current Biology*) and more explicitly spell out what new insights this study brings given such.

2. The main goal of the study is to show a common spatial representation of the transitive structures across three content types. However, a fundamental issue is that there is a big gap between the behavioral results (which shows that subjects acquired the relationship to be learned) and the neural data (which shows that activities in certain brain regions demonstrated patterns that resemble the predictions from a spatial model to some degree). In particular, there is no information about what strategies the subjects might be using or how it pertains to the nature of the underlying representation. For instance, what if subjects form explicit associative/verbal memories on the ranking of each item (e.g. Dylan – 1, Kevin – 2, William – 3)? In that case, subjects just need to activate such memory in order to do the hierarchy probe, and the RSA result might come from differences in numerical values of the rankings, which would require no spatial model. Alternatively, if the authors think that their account is agnostic about the intermediate cognitive level, they should clearly explain why.

3. A lot of details of the RSA employed are unclear/confusing, which makes the evaluation of the validity of the results difficult:

- In Line 209, it is stated that the RSA was conducted on the group level using the average of the individual subjects. At what stage was the average taken? Did the authors average the raw BOLD activities, the per-voxel betas, or the correlation/(dis)similarity matrices across subjects?
- In Line 218, how are the z-values in the z-maps calculated? Typically, a (second-order) correlation between the lower or upper triangles of the empirical RDM and the model RDM is calculated, which is not a z-value. If the authors employed any transformations, they should be clearly specified.
- In Line 254-255, in order to look for regions that showed significant RSA clusters across content types, the authors averaged the z-maps for the three conditions. Why not use 3-way overlaps (conjunctions) instead? I may be missing something, but it seems to me that the average is not a good metric for "activations" across all conditions, as a high average z-value could be driven by one or two particularly strong conditions.
- In Line 510, it is mentioned that the pre-processing of the fMRI data involves spatial smoothing with a 5mm FWHM Gaussian kernel. I might not be keeping up with the most up-to-date consensus of best practices, but I am under the impression that spatial smoothing is usually not recommended for RSA (or other MVPA analysis) as it might reduce the sensitivity to the multi-voxel patterns (Chikazoe et al., 2014, *Nature Neuroscience*; Kamitani et al., 2005, *Nature Neuroscience*).

4. I am not convinced that the NeuroSynth result provides much support on the spatial nature of the representation, although it might be a weak piece of information that is in favor of the authors' hypothesis. This seems to me an example of reverse inference, which is generally avoided and only used with extreme caution. I would recommend that the authors tone down their interpretations and make the caveats much more clear.

We would like to thank all of the reviewers for their comments and helpful feedback. We have made several changes based on your reviews, focusing on clarifying the advancements made in this study, improving the descriptions of the methodology, fine-tuning interpretations of our results, and modifying language used in the manuscript to be more specific. With these changes, we believe the manuscript has been significantly improved. Responses to specific comments can be found below (sorted by reviewer).

Sincerely,
Katherine Alfred, Andrew Connolly, Joshua Cetron, and David Kraemer

Reviewer #1 (Remarks to the Author):

This multivariate fMRI study tests the hypothesis that the mental models that we construct in order to represent hierarchies among individual items, which we can then draw on to perform transitive inference, are inherently spatial. It builds nicely on a prior study in which the authors showed - using representational similarity analysis, or RSA - that right inferior parietal sulcus (IPS), a region strongly implicated in spatial representations, shows a pattern of activation consistent with the representation of an ordered hierarchy of stimuli. In that initial study, the learned comparisons among stimuli (the relative height of two individuals) were inherently spatial. The present study expands on that work by teaching participants two non-spatial hierarchies in addition to the spatial one. Once again, right IPS appeared as a key focus in the RSA analysis. Other clusters were also observed, including one in frontopolar cortex, but only the right IPS showed overlap with the Neurosynth meta-analysis for the term 'spatial'. This is a fabulous study. I have only two comments:

1) The authors note that some comparisons are more readily 'spatializeable' than others. As such, it would be nice to see whether right IPS shows a gradient in strength of representational similarity for height > time > the abstract quality ('vilchiness').

We would like to thank the reviewer for their encouraging and helpful comments. We checked the strength of representational similarity in the right IPS for each of the three content conditions (height, price, and abstract). We found that the right IPS, regardless of condition, showed very similar levels of representational similarity (average ROI permuted Z of 2.3 for Height, 2.5 for Price, and 2.6 for Abstract). We believe that this pattern emerges because even though there is a gradient of how inherently spatial the content is for each condition (with height being more spatial than price, which is more spatial than abstract), we predict that the right IPS is actually encoding the spatial structure of the problem. A gradient of strength of association (height > price > abstract) would indicate that the right IPS is responsive to the content of the problem (in part or in entirety). Instead, this very consistent level of the strength of association indicates that it really is the spatial structure of the mental models created through transitive reasoning problems that accounts for this pattern of activity in the right IPS. We think this is an

excellent point to bring up, and we have added a short section to discuss this in the results section (Page 16):

It is important to note that while the content in the height condition is more spatial than the price condition, which is more spatial than the abstract condition, we did not see any significant differences in the average Z-values in the right IPS between conditions, indicating that the right IPS had a similar strength of representational similarity with the model in each of the conditions, regardless of problem content. This supports our hypothesis that the patterns of activity in the right IPS better represent information that is common between all the problem content types. In short- the right IPS seems to be representing information about the common structure of the mental model constructed through transitive reasoning rather than the content of the specific transitive reasoning problems.

2) The final analysis reveals a cluster in frontopolar cortex; it is quite ventral and anterior to the region in anterior prefrontal cortex that has been implicated in univariate fMRI studies of transitive inference and other reasoning tasks (often referred to as rostralateral prefrontal cortex). This is worth noting explicitly. Perhaps rostralateral PFC is involved in the process of actively drawing comparisons on the fly, but not in drawing on representations from a learned hierarchy.

We agree that this is an excellent point to note, and we have now included in the revised manuscript several sentences to that effect in our results section discussing the average content RSA results (Page 19):

Whereas a specific location in the rostralateral prefrontal cortex (RLPFC) is frequently reported in studies of transitive reasoning (Bunge et al., 2009; Waechter et al., 2013; Wendelken et al., 2008), we did not find that same region of the RLPFC to be significantly correlated with the average RSA for rank across content types. Instead, we find a cluster that is anterior and ventral to the typically reported region. Notably, these previous studies used tasks that differed from the current task in that participants were drawing new transitive inferences during the collection of neural data rather than querying a mental model previously created through transitive reasoning. Therefore, the different PFC regions implicated by the current task may be related to the process of querying a mental model previously created through transitive reasoning processes.

Reviewer #2 (Remarks to the Author):

The manuscript reports an experiment investigating the neural basis of transitive reasoning, when the type of ordinal relationship, and accompanying visual cues, are varied. The main finding is that across three “content” types transitive reasoning recruited some of the same brain regions, specifically IPS, which is seemingly involved in spatial reasoning. The authors conclude that, even when the ordinal relationship that is being reasoned about is not spatial, brain regions involved in spatial reasoning are still recruited.

Overall, I think this is an interesting, and well-designed study (e.g. the statistical analyses, in particular the permutation test, are appropriate given the experimental

design), and the main finding is compelling. However I do have some concerns primarily about how the experimental design is characterized and the results are interpreted.

Main comments:

A. [...]

The point I am trying to get to is this: it seems that ordinal relationships are ones that we naturally think of in spatial terms, and in that respect, all three tasks involve reasoning about relations that are potentially equally “spatial” in nature. In which case, it would make sense that they all recruit the same spatial brain areas, because transitive reasoning is a kind of relationship that maps nicely onto spatial relationships. The authors acknowledge this in the discussion when they say that “the involvement of [SPL and aPFC] across a spectrum of spatial and non-spatial content indicates that the involvement of the IPS is due to the spatial structure of the representational space and not the spatial content of the problems” (L323-325). But I think it would make sense to acknowledge this idea from the outset. More specifically, the authors could articulate explicitly two alternatives: (1) that if what matters is the spatial content, then we would expect to see differential activity in the price and abstract tasks (with the latter recruiting no spatial regions); and (2) if it is the spatial structure, we would expect to see similar spatial processing brain areas light up across tasks. I think being clearer about these two alternatives might help to clarify the novelty of the study.

The reviewer is completely correct that the two alternative hypotheses presented (whether the spatial activity is due to content and the SPL will decrease in activity across conditions, or whether the SPL is equally involved in all conditions due to the spatial organization of all types of content) are the crux of the argument for this manuscript, and should be made explicit much earlier in the manuscript than it currently is. We have modified the second paragraph of the introduction to clarify our primary argument (Page 5):

Specifically, previous studies have been unable to disambiguate between two competing explanations for the SPL involvement in transitive reasoning processes. It is possible that the SPL involvement is due to the use of spatial content in the reasoning tasks (such as asking participants to judge height, size, or other visually perceivable features) and that transitive reasoning tasks with completely abstract content would result in reduced or absent SPL involvement. Alternatively, it is possible that transitive reasoning itself is a highly spatial process regardless of the content of the specific problems (organizing items in the problem based on relative position to each other in the problem space), and that individuals construct a spatial mental model structure to contain and manipulate information obtained through transitive reasoning problems.

B. The talk of “content” is a bit obscure. What is clearly meant is the “representational content”, or what it is that is being represented when subjects are doing the transitive reasoning. However, per my previous comment, the content that subjects are really

reasoning about is the ordinal structure, and that is identical across tasks. So I think it would be useful to disambiguate that there are two kinds of contents here: the subject matter, and the ordinal structure. It might also help readers (who are less used to seeing 'content' being used in a technical sense) to have a sentence defining what is meant when referring to task "content". Related to this is what is meant by "mental model" and whether it is spatial.

The difference between spatial content and spatial structure is certainly a nuanced distinction, and the reviewer is correct to point out that it requires further elaboration for readers to fully understand the rest of the manuscript. We have added a paragraph that goes into more detail and provides some examples of what it would mean for a transitive reasoning problem to have spatial content compared to spatial structure (Pages 5-6):

It is important to note the difference between what this paper will refer to as "content" versus "structure" in the context of transitive reasoning problems. "Content" in this article will refer to the domain of the specific problem materials. For example, a problem may have spatial content if participants reason that one person is taller than another, or if a person is ahead of another person in line. These examples contain spatial content because there is bottom-up spatial information that is visually perceivable to the participant. This inherent spatial information allows the participant to solve the problem through perception of existing spatial relationships instead of through reasoning alone. In contrast, a problem might not have spatial content if an individual is reasoning about how expensive one painting is compared to another painting or ordering a set of objects with an arbitrary ranking dimension. This use of spatial content is contrasted with the idea of the spatial "structure" of a mental model of the problem space. A problem might be solvable by generating a mental model with meaningful spatial structure if an individual is organizing components of a problem in a spatial way regardless of the specific items specified in the transitive reasoning problem. For example, each of the types of content described above, whether spatial or non-spatial in content, could be organized in a spatial arrangement from "most" on the left to "least" on the right.

C. It seems one could draw a connection between these results and the issue of embodiment and the format of the representations that underlie transitive reasoning. If space permits in the discussion, it might be useful if the authors said a little bit about what implications these results have for that debate/topic (e.g. in relation to the final paragraph of the discussion, where transitive reasoning is contrasted with other kinds of reasoning, which might not require mental models).

We agree that is valuable that our results are able to speak to the ongoing discussion on embodied cognition. Although the topic of embodied cognition encompasses many different types of theories and sub-topics, we think that the area most directly related to our current results is that regarding Dehaene et al.'s (2007) "Neuronal Recycling Hypothesis," and related ideas. We have added a few sentences that touch briefly on the topic to the final paragraph of the discussion (Pages 24-25):

While it is meaningful to show the role of the superior parietal cortex in the creation and support of mental models created through transitive reasoning tasks, this finding has some limitations. Notably, transitive reasoning is a special kind of reasoning that differs both from other types of deductive reasoning (such as set inclusion problems), and also from inductive reasoning. A large body of work has shown that transitive reasoning is likely supported by mental models, which is not necessarily the case for all other types of reasoning. Because transitive reasoning is a form of reasoning that specifically compares relative magnitudes of items, it is especially likely that this form of reasoning utilizes the same mechanisms that allow individuals to compare directly perceived differences in magnitude. Parkinson and colleagues (2014) identified and tested this link by finding that literal spatial distance (near and far from an individual), temporal distance (near and far in time from the present moment), and social distance (close friend or stranger to an individual) shared common spatial magnitude processing in the right inferior parietal lobule. This finding of shared processing for literal and abstract distance is in line with theories put forth by Dehaene and Cohen (2007) that proposed that complex human cognition, such as the processes that support mathematics and reasoning, are supported by existing neural circuits that performed the most similar functions to the new complex cognitive functions. Because mental models tend to be represented spatially, including as mental maps of a given problem space, the information contained in mental models created through transitive reasoning seems to frequently be represented in neural regions that encode literal spatial maps or spatial distance, such as the hippocampal place maps (Frank et al., 2003; Heckers et al., 2004; Van Opstal et al., 2008; Van Opstal et al., 2009; Zalesak & Heckers, 2009; Constantinescu et al., 2016; Garvert et al., 2017; Schafer & Schiller, 2017; Theves et al., 2019) and the right parietal cortex, especially the right IPS (Krawczyk, 2012; Waechter et al., 2013; Vendetti & Bunge, 2014; Parkinson et al., 2014; Wendelken 2015; Schafer & Schiller, 2017; Alfred et al., 2018). Although there were a variety of initial content types in each of the transitive reasoning problems, representations of the content of those problems converged in a region that represented the common information between the content types- a shared spatial representation of relative magnitudes of distance. Ultimately, by systematically varying the spatial format of the content, this approach allows us to directly test whether the spatial aPFC and SPL representations of mental models created through transitive reasoning are spatial due to the content or structure and we conclude that the mental models themselves are likely spatial.

D. I think a few more methodological details (see minor comments below about the searchlight, training task) should be mentioned in the results. I didn't quite understand the findings of the experiment until I had read through the methods as well, and then went back and read the results again.

We have revamped the method and results section to include a lot of the originally missing or hard to find details. Specific responses to individual minor points can be found below, and a fully re-written method subsection about the RSA can be seen in point 14 below.

Minor Comments

1. L51, unclear what is meant by "basic information".

Updated line 51:

These representations can be built up through direct experience of a situation or they can be inferred from sparse abstract content.

2. L72-79, the reference to the “approach” here is strangely opaque. It should be described here explicitly.

Clarified and explicitly described approach (Page 6):

Instead, neural patterns representing inferred mental models can be better examined using representational similarity analysis (RSA) to allow researchers to directly measure the representation of the informational content of the reasoning problem, even when the participant is no longer viewing stimuli demonstrating the transitive relationship (as used by Alfred et al., 2018). This approach requires neural data from participants while they are considering each item in a problem space one at a time, but while thinking about how that item compares to the other items in the problem. Then, a dissimilarity matrix is calculated through how the pattern of neural activity for each item differs from each of the other items. The pattern of neural activity associated with each of the items is then correlated to a second dissimilarity matrix, which is created with how the items would theoretically differ from each other, if the participant solved the problem correctly. Locating regions of the brain where the correlation between the neural dissimilarity matrix and the modeled dissimilarity matrix is high indicates that the pattern we would expect to find is being represented by the relative patterns of neural activity in that region.

3. L73, the embedded clause “which leverages...analysis” should probably have commas at the start and finish.

Sentence has been fully reworked, removing the embedded clause (see above).

4. L110, what does “the region” refer to? SPL or IPS? Please disambiguate. I assume the authors mean IPS.

Clarified that the sentence intended to refer to the broader SPL region, including the IPS:

As noted above, the SPL (including the IPS) is also associated with transitive reasoning tasks and mental models using spatial content and perceived spatial relationships.

5. L121, sentence is unclear. What does “it” refer to in “it is associated”? SPL not V2 I assume?

Clarified the referent in L121 to indicate “it” refers to V2:

Whereas patterns of neural activity during transitive reasoning problems all show SPL activity regardless of whether the content is easy or difficult to visualize, only problems with content that

is easy to visualize show V2 activity, and that additional V2 activity is associated with relatively slower response times (Knauff et al., 2003).

6. L142-145, here is a case where one could simply refer to spatial “subject matter” rather than “content”, if one was inclined to drop the content-task.

Clarified that the task used in the former study was a spatial task (clarifying what “content” refers to):

However, the study was limited by the use of an inherently spatial task (relative height), making it more difficult to determine that the structure of the representation was spatial, *per se*, since the content was spatial as well.

7. L208, from context one can determine the authors carried out a surface-based RSA searchlight, however as far as I can tell the authors do not explicitly state they are carrying out a searchlight analysis until the methods. Since results are coming before methods, a bit more details are needed here to describe exactly what the analysis is.

Included more detail from the Method section in the Results section to better support the Results-first structure. For example, here is the paragraph in the Results that discusses the searchlight approach (Page 14):

A 100-voxel surface-based searchlight RSA (Oosterhof et al., 2011) was conducted on the group level (created using the average untargeted neural dissimilarity matrix of the individual subjects) that reflected the Pearson correlation between local neural representational structure and a target similarity structure for each of the content conditions (Height, Price, and Abstract; Figure 2). At each searchlight location, the local neural dissimilarity matrix was computed using correlation distance between activity patterns (derived from betas in the item-level GLM, described in detail in Method section) for all pairs of stimuli within that content type. Activity patterns were defined by the voxel-wise estimated hemodynamic responses from GLM analysis of the functional data collected during the two Hierarchy Probe runs for each of the three content conditions (excluding the portion of the trial in which a button response occurred).

8. I would maybe consider combining Figure 5 with 4, and just show the NeuroSynth patches on the latter, since Figure 5 is just repeating a portion of 5, as far as I can tell.

Condensed Figures 4 and 5 into a larger version of Figure 4 with Figure 5 being new Figure 4B.

Figure 4. A. Common brain regions representing mental models across content types. For all content-specific z-maps (Figure 3), values below the permutation-calculated noise threshold were removed. The three content-specific z-maps were then averaged (including 0 values for regions that did not reach significance in permutation correcting) to create the content-average map representing the common mental model structure across content types. Therefore, the z-values in this figure are averages of the permutation-corrected z-values from the content-specific maps (and as a result this scale is lower than that of Fig. 3). The average z-map was then thresholded using a bootstrapped cluster significance level of $p < .05$, corrected (minimum area = 135 mm^2), so only values within significant clusters are displayed. The full cluster list can be seen in Table 1. B. Overlap of spatial network and mental model representations. Average of permutation-corrected content RSA z-maps overlaid on top of term-based automated meta-analysis generated using NeuroSynth (“spatial” association map). Average z-map was clusterized with cluster size threshold significant at $p < .05$ (minimum area = 135 mm^2). The indicated right IPS cluster is the only cluster from the previous analysis that overlaps with the NeuroSynth “spatial” association map.

9. L357, typo. Should be either “differs” or “is different”.

Thanks for catching that! Updated to “differs”.

First, transitive reasoning is a special kind of reasoning that differs both from other types of deductive reasoning (such as set inclusion problems), and also from inductive reasoning.

10. L372, is this a typo? Should “materials and methods” be deleted?

Apologies, this is a carryover marker from an earlier version! It has been removed.

11. L390, I think this is the first place it is mentioned that subjects only saw adjacent pairs during training. If so, this is an important methodological detail that should be mentioned in the results.

We have added a small section to the results and a few sentences in the introduction that moves up some information from the method section describing that (and why) participants were only presented with adjacent pairs during training:

Page 9: These direct comparisons of adjacent items in the hierarchy (“Dylan is taller than Kevin”) were chained together to create a linear space, where the 12 face-name pairs could be ranked from tallest to shortest.

Page 10: As in the previous study, only adjacent pairs of stimuli were ever presented. Participants needed to use transitive reasoning to infer the more distal relationships that are needed to construct the full structure of the accurate mental model.

Page 11: Because the traditional format of transitive reasoning tasks takes the form of “A is more than B, B is more than C, Therefore, A is more than C”, participants were only exposed to adjacent pairs during training. The transitive reasoning comes from inferring the relationship between A and C across their shared partner, B. This aspect of transitive inference can then be tested to determine if participants reasoned correctly by querying participants for both the full ordered list (hierarchy) as well as asking about how specific pairs of items relate to each other (forced choice pairs, below). Participants were not given any feedback during any training session about if their inferred ordering was correct.

12. Why only have 3 options in the hierarchy probe when there are 5 items? Was this because of scanner limitations for performing the task in the scanner?

Only allowing 3 response options for the 5 items is partially a carryover from the original version of the study (in order to make the results from the “height” condition of this study as comparable as possible with the results from the prior study). Further, due to scanner button box limitations, we were only able to include a maximum of 4 buttons, and we decided that 3 categories (more, average, and less) was a more valid description of relative positions compared to four. We agree that it is a little strange to have 3 button options for 5 rankings, and the specific button selections from that task was not analyzed. Rather, we used that task to get participants to actively consider the placement of each item in the space relative to the other items for the RSA analysis. We instead used two tasks to determine accuracy (Forced-Choice Pairs and Hierarchy Reconstruction).

13. Figure 7, there is a lot of redundant text in this figure when depicting the training sessions. Perhaps the authors could try and simplify things. I would also include a vertical arrow to indicate that boxes at each stage are ordered by time vertically (e.g. for the fMRI session).

We have altered Figure 7 (now Figure 6 due to condensing Figures 4+5) to remove redundant information and indicate chronological order within sessions as well as between sessions. We have moved that information into the figure caption so the figure itself is less cluttered and easier to follow:

Figure 6. Experimental session timeline. All participants underwent two half-hour training sessions where they were trained on the order of the items in each content domain. Within training session each content type block, participants completed a set of tasks in a fixed order to learn the hierarchy and practice the tasks they would need to do in the fMRI session. In each of the blocks in the Training Sessions, participants first observed pairwise comparisons of only adjacent pairs for each content type (e.g. comparing paintings 1 and 2, and not paintings 1 and 4). Participants then completed the Hierarchy Probe, Forced Choice Pairs, and the Hierarchy Recall before proceeding to the next content block. The ordering of these tasks within each content block in the training sessions was always in this set order. Both of the training sessions were 24-48 hours apart and the second training session was 24-48 hours before the fMRI session. In the fMRI session, participants completed all learning review during anatomical scans at the beginning. In each content block, participants completed the Hierarchy Probe and Forced Choice Pairs tasks. Out of the scanner, participants completed the Hierarchy Recall for all content types and the Post Experimental Questionnaire.

14. L526-541, some queries about RSA. Did the authors construct neural similarity or dissimilarity matrices? In the text it is a bit ambiguous. And why did the authors calculate the Pearson's r coefficient between behavioral and neural RDMs, since the rank-order correlation (Spearman's ρ or Kendall's τ) is used, since they do not assume linearity, only monotonicity (see. e.g. Kriegeskorte, Mur, and Bandettini, 2008)?

- A) We have now clarified in the Method section that we constructed neural dissimilarity matrices (using PyMVPA's PDist function), and we have further

clarified additional details about how the content-specific RSAs were calculated as well as the rank average RSA (Pages 33-35).

Representational Similarity Analysis: The following analyses were performed using Python and PyMVPA (<http://www.pymvpa.org>; Hanke et al., 2009), SciPy (<http://scipy.org>), and NumPy (<http://numpy.scipy.org>). The searchlight-based representational similarity analysis (RSA) was conducted on the neural surface, using a 100 voxel searchlight mapping technique (Oosterhof et al., 2011) that produced a whole-brain map on the group level (created using the average of the individual subjects untargeted neural similarity for each content condition) that reflected the Pearson correlation between local neural representational structure and a target similarity structure for each of the content conditions (Height, Price, and Abstract; Figure 2). Each modeled dissimilarity matrix was created using the ordinal ranking of the objects, where the tallest/most expensive/“vilchiest” object is one distance away from the second object, two from the third object, and so on for all five items in each hierarchy. At each searchlight location, the local neural dissimilarity matrix was computed using correlation distance between activity patterns (derived from beta values in the item-level GLM, described above) for all pairs of stimuli within that content type. Activity patterns were defined by the voxel-wise estimated hemodynamic responses from GLM analysis of the functional data collected during the two Hierarchy Probe runs for each of the three content conditions (excluding the portion of the trial in which a button response occurred). After all individual untargeted neural dissimilarity matrices were calculated, a group average was calculated within each content type. Each of these content-specific average neural maps were then correlated with the modeled dissimilarity matrices for item rankings within each content type. The resulting correlations were permutation-corrected against a null distribution of 10,000 randomized potential correlation maps, resulting in a corrected Z-map for each content type (Figure 3) indicating the likelihood that any observed correlation was significantly different from chance. For further details, please see the section on multiple comparison correction below.

These three content-specific maps were further used as input for the higher-level Rank Average RSA analysis. In this analysis, we aimed to identify neural regions that showed patterns of activity that matched the predicted structure for the mental models across multiple content types. We chose to average the content-specific permutation-corrected z-maps rather than find the conjunction of all three content-specific maps because we were concerned about the potential rate of Type II error coming from only considering results that had passed several levels of strict corrections. By using averages, we would allow data from regions that were above the noise threshold and reached a trend level, but did not independently reach significance, to help identify areas that are significant when all of the content maps are considered at the same time. Through our generated distributions from our permutation corrections (described in detail below), we identified that the noise threshold was $z > 1.36$. Given we were testing our hypothesis about a positive correlation between the theoretical mental model and the pattern of neural activity, we set our cutoff for the threshold for minimum values from each of the permutation-corrected content-specific maps as $z > 1.65$, a one-tailed $p < .05$. In short, when the average rank z-map was calculated, only permutation-corrected z values greater than 1.65 from each of the content-specific RSAs were included. All other values were set to 0 to prevent averages

being artificially inflated due to noise. After the content-specific permutation-corrected z-maps that were thresholded at $z > 1.65$ were averaged together to create one average map that represented relative rank of items, across content types, we used AFNI's surface-based cluster simulation to identify significant clusters on the surface (clusters would be required to have an area greater than 135 mm²).

After clusters were identified, we further verified that the clusters we found that we had also hypothesized we would find were not due to an exceptionally high value from a single content specific map. To do so, we identified that the maximum permutation-corrected z that was included in the average was 4.1. Given the average map was calculated from three maps, an average between 4.1 and two additional 0s (non-significant values from the other two maps were masked as 0) resulted in a z of 1.36. This maximum value is the highest possible value that could have resulted from a single map contributing to a region in the average map. The cluster in the right IPS and the left aPFC both had peak and cluster-average z values higher than that cutoff, indicating that both of those clusters could not have possibly been present due to the results from a single content-specific map, and minimally indicate that multiple types of content show the same pattern of activity in those regions.

- B) In short, we used Pearson's r primarily due to practices standard in our lab and collaborators, based on the frequent usage of Pearson's r in RSA and MVPA literature (e.g. Kriegeskorte et al., 2008; Haxby et al., 2014). As succinctly stated in Haxby et al., 2014, Pearson's r is typically used because, "Measures of angular similarity such as cosine and Pearson product-moment correlation are standard measures that are most sensitive to the relative contributions of feature dimensions."

Kriegeskorte, N., Mur, M., & Bandettini, P. A. (2008). Representational similarity analysis-connecting the branches of systems neuroscience. *Frontiers in systems neuroscience*, 2, 4.

Haxby, J. V., Connolly, A. C., & Guntupalli, J. S. (2014). Decoding neural representational spaces using multivariate pattern analysis. *Annual review of neuroscience*, 37, 435-456.

Reviewer #3 (Remarks to the Author):

[...]

In particular, what contributions this study brings are a bit unclear to me, and it is also hard to evaluate how much the current results support the main conclusions because of the lack of clarity in the results and certain controls for alternative hypotheses. My detailed comments are as follows.

We thank the reviewer for this feedback, and hope that we clarified and addressed this through our responses detailed below and throughout this letter.

1. *“Mental model” is a very broad term, which could mean a lot of different things in different fields. Given the rather specific type of mental model that the task requires, it is necessary for the authors to tone down the claims made in the title and throughout the article (esp. Introduction and Discussion) to something more precise such as “transitive structure” or “relational inference”. More work also needs to be done to improve the writing of the Introduction and better position this study in the context of extant literature. Specifically, considering the broad audience of this journal, it will be helpful for the authors to mention existing studies showing spatial coding of fairly complicated non-spatial mental models already (e.g. Constantinescu et al., Science, 2016; Garvert et al., eLife, 2017; Schafer & Schiller, 2017, Neuron; Theves et al., 2019, Current Biology) and more explicitly spell out what new insights this study brings given such.*

Editor willing, we will revise the title to reflect this point: “The Neural Representation of Mental Models Used for Reasoning Across Content Type: A Common Spatial Structure”

Further, we agree that the manuscript was previously lacking a discussion of the critical role that the hippocampus plays in the creation of spatial mental maps and mental models in support of relational transitive reasoning (and that the prior research has investigated the role of the hippocampus in those informational structures using complex non-spatial information). We have added a paragraph to properly address this prior literature, where this study stands in relation to it, and why we have chosen to focus on the role of the parietal and prefrontal cortex in this study (Pages 3-4):

The neural representation of transitive and relational reasoning has frequently tested the prediction that relational information is presented in a spatial way, either through the creation of a mental map or a mental model. These representation of mental models of spatial relational reasoning has typically been investigated in hippocampal and entorhinal cortex (Frank et al., 2003 in rats; Heckers et al., 2004; Van Opstal et al., 2008; Van Opstal et al., 2009; Zalesak & Heckers, 2009; Constantinescu et al., 2016; Garvert et al., 2017; Schafer & Schiller, 2017; Theves et al., 2019). Although the involvement of the hippocampus in the support of mental maps of relational information is a critical finding (and has often been demonstrated using abstract non-spatial relationships), there is evidence that the hippocampus is not the sole region responsible for this process. Some studies (such as Wendelken et al., 2009; Schafer & Schiller, 2017) have identified the involvement of SPL and RLPFC alongside the hippocampus with regards to the encoding of mental models created through transitive reasoning. Specifically, in Wendelken et al., 2009, the authors noted that the hippocampus was specifically implicated in the process of drawing a new transitive inference across learned previously-associated pairs. In this study, we are primarily interested in testing a specific prediction of mental model theory-specifically, that mental models are separately created and queried (Johnson-Laird, 2010). To test this prediction, we separate the initial transitive inference from asking participants for how each item relates to the others in the broader space. Because this study is not collecting neural

data at the time of the initial transitive inference, based on the finding from Wendelken et al., 2009, we have chosen to focus on the role of the parietal and prefrontal cortex in this study.

2. The main goal of the study is to show a common spatial representation of the transitive structures across three content types. However, a fundamental issue is that there is a big gap between the behavioral results (which shows that subjects acquired the relationship to be learned) and the neural data (which shows that activities in certain brain regions demonstrated patterns that resemble the predictions from a spatial model to some degree). In particular, there is no information about what strategies the subjects might be using or how it pertains to the nature of the underlying representation. For instance, what if subjects form explicit associative/verbal memories on the ranking of each item (e.g. Dylan – 1, Kevin – 2, William – 3)? In that case, subjects just need to activate such memory in order to do the hierarchy probe, and the RSA result might come from differences in numerical values of the rankings, which would require no spatial model. Alternatively, if the authors think that their account is agnostic about the intermediate cognitive level, they should clearly explain why.

We agree that it is very important to understand what strategies participants were engaging in during the task, which is why we collected this information during a debriefing questionnaire immediately following the fMRI scan. That (largely qualitative) dataset had not made it to the original manuscript, but we have added it back in as it does speak to what processes individuals are engaging while undergoing this task. We have added a short section describing the questionnaire to the method section, and describe the results in the behavioral results section (Page 12):

During the debriefing questionnaire after the fMRI session, participants were asked how they were thinking about the people, paintings, and objects and how each of those items related to all the other items in the space. The overwhelming majority (all except two) described using an explicitly spatial strategy, such as putting the items in the line, imagining them on a timeline, or creating a hierarchy. Of the two participants who did not use an explicitly spatial strategy, one of the participants reported numbering each of the items, and the other participant created a mental list using the first letter from each of the words. These results indicate that our transitive reasoning problems seem to inherently encourage the use of spatial structures to organize information.

3. A lot of details of the RSA employed are unclear/confusing, which makes the evaluation of the validity of the results difficult:

- In Line 209, it is stated that the RSA was conducted on the group level using the average of the individual subjects. At what stage was the average taken? Did the authors average the raw BOLD activities, the per-voxel betas, or the correlation/(dis)similarity matrices across subjects?

- In Line 218, how are the z-values in the z-maps calculated? Typically, a (second-order) correlation between the lower or upper triangles of the empirical RDM and the

model RDM is calculated, which is not a z-value. If the authors employed any transformations, they should be clearly specified.

- In Line 254-255, in order to look for regions that showed significant RSA clusters across content types, the authors averaged the z-maps for the three conditions. Why not use 3-way overlaps (conjunctions) instead? I may be missing something, but it seems to me that the average is not a good metric for “activations” across all conditions, as a high average z-value could be driven by one or two particularly strong conditions.

- In Line 510, it is mentioned that the pre-processing of the fMRI data involves spatial smoothing with a 5mm FWHM Gaussian kernel. I might not be keeping up with the most up-to-date consensus of best practices, but I am under the impression that spatial smoothing is usually not recommended for RSA (or other MVPA analysis) as it might reduce the sensitivity to the multi-voxel patterns (Chikazoe et al., 2014, Nature Neuroscience; Kamitani et al., 2005, Nature Neuroscience).

We thank the reviewer for the request for additional information about the RSA, calculation of z-values, and other details. We received a similar request from reviewer 2, and we have extensively updated the Method and Results section to clarify these points, and here we have excerpted two relevant sections, first from the Results, second from the Method:

From the Results (Pages 13-14):

Neural Representational Similarity Analysis (RSA)

A 100-voxel surface-based searchlight RSA (Oosterhof et al., 2011) was conducted on the group level (created using the average untargeted neural dissimilarity matrix of the individual subjects) that reflected the Pearson correlation between local neural representational structure and a target similarity structure for each of the content conditions (Height, Price, and Abstract; Figure 2A). At each searchlight location, the local neural dissimilarity matrix was computed using correlation distance between activity patterns (derived from betas in the item-level GLM, described in detail in Method section) for all pairs of stimuli within that content type. Activity patterns were defined by the voxel-wise estimated hemodynamic responses from GLM analysis of the functional data collected during the two Hierarchy Probe runs for each of the three content conditions (excluding the portion of the trial in which a button response occurred). The RSA resulted in a correlation value for each surface node based on how closely the pattern of neural activity in that region mirrored the pattern of the mental model given in the dissimilarity matrix, for each of the three content types (Figure 3). To correct for multiple comparisons, we conducted a permutation test by shuffling the labels on our data 10,000 times to create a distribution for our data. We then z-scored the correlation with the *a priori* model at each node to this distribution created from our data to find how likely it is that our results occurred by chance (Figure 2B).. The complete results of the permuted z-maps (where $p < .01$) for each of the content types can be seen in Figure 3. For further correction of multiple comparisons, we corrected the permuted z-maps at the cluster level to retain only spatially contiguous surface nodes that exceed the likelihood threshold indicating they were not likely to have been identified by chance (see Method section for full details).

From the Method (Pages 33-35):

Representational Similarity Analysis: The following analyses were performed using Python and PyMVPA (<http://www.pympva.org>; Hanke et al., 2009), SciPy (<http://scipy.org>), and NumPy (<http://numpy.scipy.org>). The searchlight-based representational similarity analysis (RSA) was conducted on the neural surface, using a 100 voxel searchlight mapping technique (Oosterhof et al., 2011) that produced a whole-brain map on the group level (created using the average of the individual subjects) that reflected the Pearson correlation between local neural representational structure and a target similarity structure for each of the content conditions (Height, Price, and Abstract; Figure 2). Each modeled dissimilarity matrix was created using the ordinal ranking of the objects, where the tallest/most expensive/“vilchiest” object is one distance away from the second object, two from the third object, and so on for all five items in each hierarchy. At each searchlight location, the local neural dissimilarity matrix was computed using correlation distance between activity patterns (derived from beta values in the item-level GLM, described above) for all pairs of stimuli within that content type. Activity patterns were defined by the voxel-wise estimated hemodynamic responses from GLM analysis of the functional data collected during the two Hierarchy Probe runs for each of the three content conditions (excluding the portion of the trial in which a button response occurred). After all individual untargeted neural dissimilarity matrices were calculated, a group average was calculated within each content type. Each of these content-specific average neural maps were then correlated with the modeled dissimilarity matrices for item rankings within each content type. The resulting correlations were permutation-corrected against a null distribution of 10,000 randomized potential correlation maps, resulting in a corrected Z-map for each content type (Figure 3) indicating the likelihood that any observed correlation was significantly different from chance. For further details, please see the section on multiple comparison correction below.

These three content-specific maps were further used as input for the higher-level Rank Average RSA analysis. In this analysis, we aimed to identify neural regions that showed patterns of activity that matched the predicted structure for the mental models across multiple content types. We chose to average the content-specific permutation-corrected z -maps rather than find the conjunction of all three content-specific maps because we were concerned about the potential rate of Type II error coming from only considering results that had passed several levels of strict corrections. By using averages, we would allow data from regions that were above the noise threshold and reached a trend level, but did not independently reach significance, to help identify areas that are significant when all of the content maps are considered at the same time. Through our generated distributions from our permutation corrections (described in detail below), we identified that the noise threshold was $z > 1.36$. Given we were testing our hypothesis about a positive correlation between the theoretical mental model and the pattern of neural activity, we set our cutoff for the threshold for minimum values from each of the permutation-corrected content-specific maps as $z > 1.65$, a one-tailed $p < .05$. In short, when the average rank z -map was calculated, only permutation-corrected z values greater than 1.65 from each of the content-specific RSAs were included. All other values were set to 0 to prevent averages being artificially inflated due to noise. After the content-specific permutation-corrected z -maps that were thresholded at $z > 1.65$ were averaged together to create one average map that represented relative rank of items, across content types, we used AFNI’s surface-based cluster simulation to identify significant clusters on the surface (clusters would be required to have an area greater than 135 mm²).

After clusters were identified, we further verified that the clusters we found that we had also hypothesized we would find were not due to an exceptionally high value from a single content specific map. To do so, we identified that the maximum permutation-corrected z that was included in the average was 4.1. Given the average map was calculated from three maps, an average between 4.1 and two additional 0s (non-significant values from the other two maps were masked as 0) resulted in a z of 1.36. This maximum value is the highest possible value that could have resulted from a single map contributing to a region in the average map. The cluster in the right IPS and the left aPFC both had peak and cluster-average z values higher than that cutoff, indicating that both of those clusters could not have possibly been present due to the results from a single content-specific map, and minimally indicate that multiple types of content show the same pattern of activity in those regions.

To the point about the conjunction analysis: When displaying the content-specific results, we only wanted to include values that were significant after permutation-corrections at a conservative threshold. A conjunction map of those results would have resulted in highly sparse overlaps, and it also discards valuable information about the strength of the relationship in different localizations (e.g. the IPS activity is slightly more ventral in the height condition compared to the abstract condition). Any area that is encoding the pattern of activity for the structure of the transitive reasoning problems across multiple content types is interesting for this study and therefore we would not recommend using an overly-conservative threshold at the stage of the content-specific maps (opting instead for a more conservative threshold at the final stage of analysis). However, we agree that using the unthresholded permutation-corrected z -maps at the content-specific stage might potentially allow z -values to be included that occurred due to chance in at least one content domain to influence the results. To address this concern, we have re-calculated the average RSA using thresholded permutation-corrected z -maps for each content-specific map. In order to determine an appropriate threshold for the content-specific maps, based on our permuted null distribution, we have estimated the level of noise in the data to be at a maximum of $z = \pm 1.37$, indicating that a threshold at this level would be sufficient to exclude spurious activation. Based on this calculation, we have chosen a slightly more conservative threshold at $z > 1.65$. This z -threshold corresponds to $p < .05$ (one-tailed), which is consistent with our goal of exploring only positive correlations between the models and patterns of neural activity. Accordingly, any value that was less than 1.65 was set to 0 before averaging the content-specific maps. Following this correction on the level of individual nodes from each content-specific map, we also applied an additional correction on the average map. This correction applied to the level of spatial clusterization and used a bootstrapped threshold of $p < .05$ generated by a Monte-Carlo simulation of minimum probable cluster size (using the AFNI program `slow_surf_clustsim`). This correction is appropriate given that we're testing a hypothesis about the spatial localization of nodes into clusters across the 3 analyses.

A further point addressing the reviewer's concern is that because that the maximum observed value occurring in any of the content maps is 4.1, the maximum possible value that could occur due to a high value in only one condition is 1.36. Through these corrections, all remaining cluster peaks are above 1.36 indicating overlap between

multiple content-specific z-maps. Further, the average value across the entire cluster for the two main clusters (in right IPS and left aPFC) are higher than 1.36 (1.39 and 1.46 respectively). This indicates that not only is the peak for these clusters above the minimum value to show that item rank is being represented across content types, but that the overall cluster average is also above this value. Ultimately, the peak and average permutation-corrected z values for the rank average map are convincing evidence that it is not simply a very high value in one map driving these findings, but a combination across content types, as we asserted in the manuscript. The text and figures have been updated to reflect this change. Notably, the main findings of IPS and aPFC are still present after these additional corrections. Here is the relevant excerpt from the Results section of the revised manuscript (Pages 17-18):

In order to examine the effect of rank, we identified regions that were present in all three of the permutation-corrected content RSA maps. Areas present across all three RSA z-maps are likely regions that represent rank order differences in all possible problem content types (as opposed to being content-specific). To determine candidate regions, we averaged the permutation-corrected z-maps for all three content types, thresholded so that all values less than $z < 1.65$ (one-tailed $p < .05$) were set to 0 to eliminate any values that could have been artificially boosted by chance through the averaging. From the distribution of possible correlations calculated during our permutation correction, we identified that the level of noise in the data to be at a maximum of $z = \pm 1.37$, indicating that a threshold at this level or greater would be sufficient to exclude spurious activation. Averaging the maps was selected rather than a conjunction map to better determine the average strength of association across all content maps (rather than just passing a bare minimum level of association across all maps). Given that the maximum observed value occurring in any of the content maps is 4.1, the maximum possible value that could occur due to a high value in only one condition is 1.36. We then identified significant clusters in the averaged z-map by using AFNI's surface-based cluster simulation to determine the minimum significant cluster area. Through these corrections, all remaining cluster peaks are above 1.36 indicating overlap between multiple content-specific z-maps. We only included clusters that were significant at corrected $p < .05$ (an area of 135 mm² or greater) to minimize noise. In the clusterized average z-map, we found a significant cluster in the predicted right IPS (Figure 4A for full map; Table 1 for cluster list).

D) We chose to use spatial smoothing for our neural data that went into the RSA analyses because not only has it been shown to not be harmful in most cases, it has been demonstrated to have beneficial effects. Previous research (Hendriks et al., 2017) has shown that moderate spatial smoothing (1-4x the voxel size) improved MVPA correlations between a slight to a large improvement, depending on the specific region. Moreover, even high amounts of spatial smoothing (more than used in our study) were shown to have a positive effect in some regions. While we are looking for relatively fine-grained patterns in our RSA, we're looking at the result of a relatively high-level process, and we wanted the signal to noise ratio to be as high as possible. Hanke and colleagues (2009) recommend similarly in a discussion on searchlight size for classification analyses. While not smoothing per se, single voxel searchlights result in the highest classification error rates (17% for the best performing) when compared to searchlights of 5-10 mm (8% was the best performing searchlight sphere with a size of

10 mm). Ultimately, the authors suggest that while 4 mm searchlights might be ideal for analyses of low level visual features, higher level cognitive processes are better served by larger searchlights which can boost signal-to-noise ratios.

Hanke, M., Halchenko, Y. O., Sederberg, P. B., Hanson, S. J., Haxby, J. V., & Pollmann, S. (2009). PyMVPA: a python toolbox for multivariate pattern analysis of fMRI data. *Neuroinformatics*, 7(1), 37-53.

Hendriks, M. H., Daniels, N., Pegado, F., & Op de Beeck, H. P. (2017). The effect of spatial smoothing on representational similarity in a simple Motor Paradigm. *Frontiers in neurology*, 8, 222.

4. I am not convinced that the NeuroSynth result provides much support on the spatial nature of the representation, although it might be a weak piece of information that is in favor of the authors' hypothesis. This seems to me an example of reverse inference, which is generally avoided and only used with extreme caution. I would recommend that the authors tone down their interpretations and make the caveats much more clear.

Reverse inference is something to be very careful about in scientific literature, and we agree that great care should be taken in using reverse inference to make claims. However, we believe that NeuroSynth is a special use case of reverse inference. In his 2011 *Neuron* paper, Russ Poldrack discusses the problems with reverse inference, and goes on to discuss how tools like NeuroSynth can be useful for formal predictions with neural data. We use what were previously called "reverse inference" maps from NeuroSynth, and are currently called "association maps" to more appropriately capture the best use of the data provided by the maps. In these maps, we have the "z-scores from a two-way ANOVA testing for the presence of a non-zero association between term use and voxel activation" (from neurosynth.org). Specifically, this information tells us where in the brain activity is present more for that given term (such as "spatial") than other terms. This approach controls for base rates between regions- a region that shows up commonly in studies of a large variety of topics will be less likely to appear, and these maps are more likely to show activity that is relatively specific to the queried term. When we use the spatial association map to understand if the activity in the region is likely spatial, we have taken the activations that are present in studies that use the term "spatial" more frequently than they are for studies that do not use that term.

Poldrack, R. A. (2011). Inferring mental states from neuroimaging data: from reverse inference to large-scale decoding. *Neuron*, 72(5), 692-697.

Yarkoni, T., Poldrack, R. A., Nichols, T. E., Van Essen, D. C., & Wager, T. D. (2011). Large-scale automated synthesis of human functional neuroimaging data. *Nature methods*, 8(8), 665.

Reviewers' comments:

Reviewer #1 (Remarks to the Author):

The authors have been very responsive to all three reviews, and I think the manuscript has improved as a result. I have no further concerns. Great study!

Best,
Silvia Bunge

Reviewer #2 (Remarks to the Author):

I thank the authors for their careful consideration of my comments on their manuscript on the neural representation of mental models that underlie transitive reasoning across spatial content-types. In the revisions, they have adequately addressed both my major comments (e.g. about clarifying the relationship between spatial content vs structure), and my minor comments. I just have a few further comments, all of which I think fairly easy to address.

Main comments:

A. The introduction, which was already rather long, is now considerably longer. I fully realize that this is a result of the authors trying to do their due diligence in their revisions (including to my comments!), but I suspect it could do with being shortened to make the motivation for the paper clearer and more concise. Though I leave it to the authors to decide whether to take up this suggestion.

B. More substantively, while the new content is fine, I found a number of spelling/grammar errors in the added text, as well as some sentences that were very difficult to parse. I have highlighted a few examples below in my minor comments. Overall the new material reads as though it was perhaps written in a bit of haste, and so could maybe use a little more copy editing and polish (to put it more stylistically in line with the rest of the text).

C. On p.12 the authors describe the strategies used by the participants based on the debriefing. I think this passage is actually quite important in relation to the mental models they used, since it strongly suggests similar spatial structure across tasks (as the authors acknowledge). I would recommend returning to these observations in the discussion when interpreting the results (e.g. in connection to the final paragraph of the discussion).

D. I noticed in the methods that now there is mention of one-tailed permutation tests, with a threshold of .05. In one or two cases, this seems to replace a reported threshold of .001 (e.g. in the description of the permutation test in the methods, and notes for Table 1). Was there a change in analysis, or did the authors not report the appropriate threshold values previously? Obviously one-tailed at .05 is the most liberal threshold one can have given standard conventions for testing significance. Would the authors find a different result if they used a two-tailed test?

Minor comments:

Please highlight edited text, rather than simply showing track changes, to indicate revisions. Track changes can be visually confusing, as when deleted figures still appear in the manuscript.

p.3: "The neural representation of transitive..." this does not appear to be a well formed sentence.

p.3-4 And the next sentence an agreement error: "These representations" not "These representation" and "have" not "has".

p.4. Delete "in rats" or reference the animal model somewhere other than the APA-style in text list of references.

p.4 "2009" should be in brackets for Wendelken et al. 2009.

p.4 "In this study, we are primarily..." awkward sentence. Why not just say: "we are primarily interested in testing the prediction of mental model theory that models are separately created and queried", rather than saying "specifically" twice in the same sentence.

p.4 "To test this prediction, we separate" when referring to the present study in other places, the authors use past tense. In which case here it should be "we separated". More generally, the authors should make sure they are being consistent in their use of tense between the old and new text.

p.6-7 It seems these details about RSA would fit better in the results. Here it probably suffices to just mention something general about correlating dissimilarity structures.

p.11 Maybe better to say "Participants completed the Hierarchy Recall task in both of two half-hour training sessions".

p.15 here the track changes make things very confusing as it appears as though the same figure occurs twice.

p.16-17 the similar amount of representational similarity the authors observe makes quite a lot of sense given the spatial structure was the same across tasks. This is a place where the results of the debriefing are quite instructive. So the authors should acknowledge this by explicitly referring to the "common spatial structure of the mental model constructed...".

p.18-19 I am not sure you need to say much more than you averaged the maps here. Some of the details mentioned here are redundant on those in the methods where they are more carefully described. By this I mean details like converting all other values to 0, why averaging was chosen over conjunction, and that the max possible value is 4.1. I realize I asked previously to foreshadow more of the details in the results, but this level of granularity does not seem necessary.

p.19 Should there be a space between "135mm" and "2 or greater"?

p.20 Should be "Instead, we found" not "we find" if tense is to remain consistent with the older text.

p.20 Clearly the mental models for the different tasks are equivalent in spatial structure, given what subjects said during debriefing.

p.23 Why is the definition of the p value changing in the notes for table 1? .05 is very different from .001. Because of track changes, I cannot tell what is going on in this table anymore.

p.27 "identified and tested this link and found..." not "by finding".

p.28 "information contained..." I think here you can explicitly refer to spatial structure. After doing such a nice job making the spatial content/structure distinction, one might as well use it here.

p.28 Missing comma between "normal vision" and "participated in this study".

p.33 Figure 6 legend: I am having difficulty parsing the first sentence. Should the beginning read "Within each content-type block of each training session, participants...".

p.33 Figure 6 legend: "In the fMRI session, participants" this is a very awkward sentence. What does "learning review" mean? That subjects just reviewed the material somehow?

p.38 "By using averages, we would allow data..." I am not sure you need this sentence to justify averaging.

p.39 was the bootstrap corrected threshold also one-tailed?

p.39 "After the content-specific..." this sentence is difficult to parse. There are some embedded clauses that are hard to recognize as such Please revise. The same is true for "After clusters were identified, we further verified..." There are three embeddings in "we further verified that the clusters we found that we had also hypothesized we would find...". These sentences could be made much clearer/simpler.

Reviewer #3 (Remarks to the Author):

This is a revised version of a manuscript I reviewed a while ago. This manuscript aims to investigate the neural encoding of the mental models required for transitive reasoning across different content types from visuospatial to abstract. In this version, the authors have addressed most of my concerns to satisfaction. In particular, the descriptions of the RSA pipeline and relevant details on statistics have been much improved. The additional discussions on existing studies of mental models, on how this study is positioned among these studies, and on what contributions it aims to achieve are also helpful.

I have one remaining concern on the authors' approach to identify brain regions that encode a similar spatial structure for mental models across content types. I had doubts about using an average z-value across the three content types and suggested using a conjunction instead. In this round of revision, the authors did not embrace this suggestion and offered some explanations justifying averaging, which I do not find convincing. To my understanding, the logic behind the question is quite straightforward – the authors should be looking for a logical "AND" between the clusters encoding each of the content types. Doing an averaging of the effect sizes (z-values) in the individual content types simply does not fulfill this, with or without the additional twists (e.g. thresholding and corrections) that were introduced in the current version. For example, a comparison of Figure 3 and 4 could easily find that there is no aPFC clusters for the "price" content type in Figure 3, while in Figure 4 aPFC is part of the "common brain regions representing mental models across content types" in the authors' own words. Putting the technical details aside, how could one region that is not even encoding the mental model in one content type (based on whatever statistical criteria the authors chose to use) suddenly becomes one that represents mental models across all content types? I still believe that conjunction is the right approach that fits the research question in this study, and that the authors need to grapple with this issue in a more meaningful way given the centrality of this question in their study.

We would like to thank the editor, Dr. Rosenthal, and the reviewers for their thorough and continued feedback. We have made several updates to the manuscript; in particular, we have included a conjunction analysis to address reviewer 3's concerns, and we have updated the results and discussion section to address those findings. Although we believe that we have fully addressed the reviewer's comments with these revisions, we are happy to make any further changes that might be requested.

Reviewer #1 (Remarks to the Author):

- *The authors have been very responsive to all three reviews, and I think the manuscript has improved as a result. I have no further concerns. Great study!*

Best,

Silvia Bunge

The authors would like to sincerely thank Dr. Bunge for her helpful feedback during the prior round of revisions, and we are thrilled we were able to address her concerns.

Reviewer #2 (Remarks to the Author):

- *I thank the authors for their careful consideration of my comments on their manuscript on the neural representation of mental models that underlie transitive reasoning across spatial content-types. In the revisions, they have adequately addressed both my major comments (e.g. about clarifying the relationship between spatial content vs structure), and my minor comments. I just have a few further comments, all of which I think fairly easy to address.*

Main comments:

- *A. The introduction, which was already rather long, is now considerably longer. I fully realize that this is a result of the authors trying to do their due diligence in their revisions (including to my comments!), but I suspect it could do with being shortened to make the motivation for the paper clearer and more concise. Though I leave it to the authors to decide whether to take up this suggestion.*

We agree that the introduction has become too long, though we appreciate the note that it became too long because we were trying to thoroughly address reviewer concerns. Ultimately,

we have cut the introduction from 2679 words to 1764 words (a little over 900 words cut). We started by finding some areas where content seemed redundant and pared that down. For example, the first two paragraphs of the introduction have had duplicate content cut and merged together to become this section of the introduction:

Mental models, cognitive architectures containing relational properties that reflect the structure of problem or situation in reality, are a critical component of human thought and reasoning (Johnson-Laird, 2010). They allow us to store information about the world, manipulate that information, and draw inferences about relationships between entities in the world. The neural representation of transitive and relational reasoning has frequently tested the prediction that relational information is presented in a spatial way, either through the creation of a mental map or a mental model. These representation of mental models of spatial relational reasoning has typically been investigated in hippocampal and entorhinal cortex (Frank et al., 2003 in rats; Heckers et al., 2004; Van Opstal et al., 2008; Van Opstal et al., 2009; Zalesak & Heckers, 2009; Constantinescu et al., 2016; Garvert et al., 2017; Schafer & Schiller, 2017; Theves et al., 2019).

Although the involvement of the hippocampus in the support of mental maps of relational information is a critical finding (and has often been demonstrated using abstract non-spatial relationships), there is evidence that the hippocampus is not the sole region responsible for this process. Some studies (such as Wendelken et al., 2009; Schafer & Schiller, 2017; for review of literature, refer to Wendelken 2015; Krawczyk, 2012; and Vendetti & Bunge, 2014) have identified the involvement of SPL and RLPFC alongside the hippocampus with regards to the encoding of mental models created through transitive reasoning. Specifically, in Wendelken et al., 2009, the authors noted that the hippocampus was specifically implicated in the process of drawing a new transitive inference across learned previously-associated pairs. In this study, we are primarily interested in testing a specific prediction of mental model theory- specifically, that mental models are separately created and queried (Johnson-Laird, 2010). We designed a paradigm that allowed us to test that prediction, in which we separated the initial transitive inference from querying participants about the relationships between items in the problem space. Because this study is not collecting neural data at the time of the initial transitive inference, based on the finding from Wendelken et al., 2009, we have chosen to focus on the role of the parietal and prefrontal cortex in this study.

We have further selected a section from the introduction (included below) which we have moved and reworked into the discussion. We think the content it adds is still valuable to the manuscript, but it does not need to be in the introduction specifically. Its modified form in the discussion is included below:

This study expands on a framework for relational reasoning put forth by Wendelken and colleagues (2010) focusing on the role of the aPFC and SPL. Whereas this framework is broadly supported by the current body of research, there is some evidence the SPL may play a greater role than simply encoding individual relationships. The result from this study indicating that only the right IPS shows significant overlap between all three content types while the aPFC fails to reach significance indicates that the role of the SPL might be larger than originally posited. Specifically, the SPL may be critical for generating transitive inference itself. In one study of

patients with aPFC or SPL lesions performing transitive reasoning tasks, only patients with SPL lesions showed significant impairment compared to aPFC lesioned patients and healthy controls; (Waechter et al., 2013). Interpreted under the framework proposed by Wendelken and colleagues (2010), lesions to either the SPL or aPFC should both have produced significant impairment to transitive reasoning. Because patients with aPFC lesions were not significantly impaired during transitive reasoning tasks, the SPL seems to be playing a critical role in both individual relationship encoding as well as generating inferences based on transitive relationships. More recent research using representational similarity analysis to probe the representation of mental models has also supported the involvement of the SPL while drawing transitive inferences (Alfred et al., 2018). Therefore, it seems plausible that the SPL is active in supporting transitive reasoning at the point of drawing inferences.

- *B. More substantively, while the new content is fine, I found a number of spelling/grammar errors in the added text, as well as some sentences that were very difficult to parse. I have highlighted a few examples below in my minor comments. Overall the new material reads as though it was perhaps written in a bit of haste, and so could maybe use a little more copy editing and polish (to put it more stylistically in line with the rest of the text).*

We apologize for the typos and grammatical errors that have arisen over the course of our revisions. We have addressed all of the errors specifically noted in the minor revisions, and we have also gone through the manuscript to rework some of the more difficult sentences to make them easier to parse. We have included some examples here as well as in the point-by-point minor comment section:

“These prior studies on the representations of mental models of spatial relational reasoning have typically been focused on investigating the role of hippocampal and entorhinal cortex (Frank et al., 2003; Heckers et al., 2004; Van Opstal et al., 2008; Van Opstal et al., 2009; Zalesak & Heckers, 2009; Constantinescu et al., 2016; Garvert et al., 2017; Schafer & Schiller, 2017; Theves et al., 2019). “

“Rather than study new transitive inference, in this study, we are primarily interested in testing the prediction of mental model theory that models are separately created and queried (Johnson-Laird, 2010).”

“We designed a paradigm that allowed us to test that prediction, in which we separated the initial transitive inference from querying participants about the relationships between items in the problem space.”

“For this study, we recruited nineteen undergraduate and graduate student participants (15 female, $M_{\text{age}} = 19.81$) who were right-handed, fluent in English, and with normal or corrected to normal vision.”

“Each content type had 5 total items resulting in 10 unique pairings per content type. The data for the forced choice pairs were collected from one run each per content type.”

- *C. On p.12 the authors describe the strategies used by the participants based on the debriefing. I think this passage is actually quite important in relation to the mental models they used, since it strongly suggests similar spatial structure across tasks (as the authors acknowledge). I would recommend returning to these observations in the discussion when interpreting the results (e.g. in connection to the final paragraph of the discussion).*

We agree that the information participants gave during debriefing is a valuable insight into how transitive reasoning problems are approached. It seems highly likely that there is something about transitive reasoning problems (and potentially other sorts of problems) that encourages the use of spatial structuring to organize information. We have called back to that information both in another location in the Results and in the discussion now as well:

From Results:

In short- the right IPS seems to be representing information about the common structure of the mental model constructed through transitive reasoning rather than the content of the specific transitive reasoning problems. All the problems used in this study were transitive reasoning problems, which have a very specific organizational structure. Consistent with our hypotheses, there is a great deal of overlap in the neural localization of mental models across content types. Further, participants overwhelmingly reported using a spatial strategy to organize the information in the transitive reasoning problems, such as organizing the information on a line. Because nearly all participants used a spatial strategy for the transitive reasoning problems in all content types, we would predict to see similar broad patterns of activity for all transitive reasoning problems unless the content itself was the primary determiner of patterns of neural activity.

From Discussion:

Although there were a variety of initial content types in each of the transitive reasoning problems, representations of the content of those problems converged in a region that represented the common information between the content types—a spatial representation of relative distances (Parkinson et al., 2012; Parkinson et al., 2014). This hypothesis is further supported by participant self-report during debriefing. Regardless of the type of content used in the transitive reasoning problems, participants stated that they used a spatial strategy to organize the information in the problems, i.e., nearly all utilized a spatial mental model.

- *D. I noticed in the methods that now there is mention of one-tailed permutation tests, with a threshold of .05. In one or two cases, this seems to replace a reported threshold of .001 (e.g. in the description of the permutation test in the methods, and notes for*

Table 1). Was there a change in analysis, or did the authors not report the appropriate threshold values previously? Obviously one-tailed at .05 is the most liberal threshold one can have given standard conventions for testing significance. Would the authors find a different result if they used a two-tailed test?

We thank the reviewer for highlighting this important point that bears further clarification. In short, all reported clusters are significant at $p < .05$, corrected (two-tailed), and the z-values for the individual nodes within a cluster are also thresholded and compared to a permuted null distribution of correlation values. In this response and in the revised manuscript, we now explain the selection of these thresholds in detail to motivate the selection of specific p-values for each analysis. To summarize, for all analyses we correct at the voxel-level through permutation corrections, in addition to correcting at a bootstrapped cluster correction for minimum significant cluster size. In analyses where the initial permutation-corrected results for individual task conditions will then be input into a second-level analysis in which they will be further corrected at the cluster level, we use a lower threshold at the first stage (to minimize type II errors), supported by data validation that demonstrates the threshold above which we are able to detect signal. Then we apply spatial cluster correction at a more conservative threshold at the second stage (to minimize type I errors). For the analyses that end after the first level (the individual task condition analyses), we use a more conservative node-level threshold (to minimize type I errors).

In the permutation correction step, we originally used a two-tailed $p < .05$ threshold, as can be seen in the individual content RSA maps, where it is still applied. Since we do not analyze the individual maps any further, we have left the significance level there. When we initially calculated the average RSA map, we did not apply the permutation corrections to the data that was used as an input for the averages, so we used a more conservative bootstrapped cluster correction threshold of $p < .001$, which can be seen in our original table of results. However, when Reviewer 3 pointed out that using uncorrected input into the average RSA would allow noise from each of the maps to combine and potentially create false signal, we agreed that we should apply some corrections to the initial RSA results before averaging. Consequently, we calculated the level of noise in our data to be $z = \pm 1.37$ (following the approach of Nili et al., 2014). That is, after generating our permuted two-tailed distribution of potential outcomes and correcting our correlations to the corrected z-values we use for the rest of the study, we determined the noise threshold for that permutation-corrected distribution was $z = 1.37$. This lets us use this value as an estimation of the level of noise in the data. We further decided on a cutoff that was even more conservative than our level of noise estimation: a permutation-corrected z-value greater than or equal to 1.65, which corresponds to the value of a one-tailed $p < .05$. Given that we were now going to be applying corrections at two stages in the analysis, we did not want to risk a high likelihood of type II error by overly-correcting our data at the initial input stage. Then, after creating the average RSA using only values higher than $z > 1.65$ (all others averaged as 0), we bootstrap cluster corrected those results to only include clusters where the spatial extent was significant at $p < .05$ (two-tailed) using input data that was already thresholded at the node level.

In this most recent version of our manuscript, we are now additionally including a conjunction analysis. This is an even more conservative analysis, as it requires a $z > 1.65$ in all content RSA maps in the same voxels (as opposed to a region significantly correlating with each content type

in slightly different sub-regions). The results of this analysis can be seen in the new inset to Figure 4, where the right IPS and left inferior frontal cortex show overlap between the three content-specific RSAs, and the surrounding region is covered by different two-content overlap (resembling a Venn diagram). For this analysis, we use the same one-tailed $p < .05$ for our permutation-corrections to the content-specific RSAs. We then calculated the bootstrapped cluster correction and only include clusters significant at two-tailed $p < .05$.

Here is the updated portion of the Results section that describes these analyses, including the corrections we described:

In order to examine neural representations of mental model structure independent of content type, we identified regions that were present across all of the individual content RSA maps. From the distribution of possible correlations calculated during our permutation correction, we identified that the level of noise in the data to be at a maximum of $z = \pm 1.37$, indicating that a threshold at this level or greater would be sufficient to exclude spurious activation. To determine candidate regions, we averaged the permutation-corrected z -maps for all three content types at a more conservative threshold than the level indicated by the noise threshold, $z > 1.65$ (one-tailed $p < .05$) to eliminate any values that could have been artificially boosted by chance through the averaging. We then identified significant clusters in the averaged z -map by using AFNI's surface-based cluster simulation to determine the minimum significant cluster area. We only included clusters that were significant at $p < .05$, corrected (an area of 120 mm² or greater). In the clusterized average z -map, we found a significant cluster in the predicted right IPS, as well as regions of prefrontal cortex (see Figure 4 for whole brain map; see Table 1 for cluster list).

Because the average z -map reveals regions that significantly correlate with the content-specific rank models combined across content types (i.e. regions where the pattern of neural activity represents the mental model of the rank order of the items), regions in this map are associated with mental models of a similar structure, rather than within a specific content type. Since the set of content types range from fully spatial (height) to not inherently spatial but spatializeable (price) to novel (vilchy), regions supporting the mental model of rank order for all these content types did not necessarily need to be spatial. It is important to note, once again, that despite the fact that the content varied from spatial to non-spatial, nearly all participants reported using spatial strategies to organize the structure of the information from these problems. Nonetheless, this analysis revealed a significant cluster in the right IPS, in a region specifically associated with estimations of spatial magnitude in previous research (Beudel et al., 2009; Silk et al., 2010). To support our assertion that the involvement of the right IPS represents spatial information specifically, we used the association map from NeuroSynth with the "spatial" keyword (NeuroSynth.org; Yarkoni et al., 2011).

This map indicates areas that are selectively active for spatial information as compared to all other terms in the database (created through meta-analysis of 1,157 studies that include "spatial" as compared to the remaining 13,214 studies, thresholded at FDR corrected .01). This approach is an objective, external, data-driven method to generate networks based on keywords in which activity across previous studies has been associated more with the term "spatial" than with other terms in the database. The spatial association map was binarized and the clusterized average z -map of the permutation-corrected and cluster-corrected RSA was overlaid on top of the spatial

association map. The region in the right IPS is the only cluster that falls within the spatial association map, indicating that the right IPS is used for creating a spatial mental model of rank order created through transitive reasoning problems, regardless of the content type used in the transitive reasoning problems (Figure 4). Whereas a specific location in the rostralateral prefrontal cortex (RLPFC) is frequently reported in studies of transitive reasoning (Bunge et al., 2009; Waechter et al., 2013; Wendelken et al., 2008), we did not find that same region of the RLPFC to be significantly correlated with the average RSA for rank across content types. Instead, we found a cluster in the left inferior frontal cortex (IFC) that responded to mental model structure across content types. This region has been previously reported in reasoning research as a region that is active during correct analogy trials compared to baseline, (Wendelken et al., 2008). The cluster in the left IFC is also seen in our previous study using only the height-based transitive reasoning task (Alfred et al., 2018). We further identified a cluster in left anterior prefrontal cortex (aPFC) that is anterior and ventral to regions identified in the aforementioned studies demonstrating RLPFC involvement in transitive reasoning. Notably, these previous studies used tasks that differed from the current task in that participants were drawing new transitive inferences during the collection of neural data rather than querying a mental model previously created through transitive reasoning. In contrast, the left IFC and aPFC regions implicated by the current task relate to the process of querying a mental model previously created through transitive reasoning processes.

Finally, to determine which regions in the average RSA analysis represented similar mental models in each of the three content types, we calculated a conjunction map. This conjunction map highlights regions where either two or three content types had significant values after the initial permutation-based thresholding step. As with the average RSA, following a node-level thresholding step ($z > 1.65$), we applied a bootstrapped spatial cluster correction to the conjunction map to only preserve clusters significant at $p < .05$, corrected. Only the clusters in the right IPS and the left IFC were both significantly large and contained overlap between all three content types. The extent of the three-way overlap in right IPS is limited, however, the area surrounding that overlap is made up of different two-content overlap sections, indicating that adjacent and partially-overlapping regions of this region are coding these spatial mental models. Similarly, the cluster in the left IFC shows sparse three-way overlap, but it is surrounded by Height-Price and Height-Abstract overlapping regions, indicating the left IFC encodes the structure of the mental model across all three content types in the same region, if not precisely the same voxels within that region. The right IPS is the only region that overlaps with the NeuroSynth “spatial” meta-analytic map, consistent with the hypothesis that this region is involved in encoding the spatial structure of mental models across all three content types. The only other region to emerge in both the conjunction analysis and the average RSA analysis is the aPFC region that shows overlap between the Height and Abstract conditions, but not Price. No other regions survived thresholding for either the conjunction analysis or the average RSA analysis.

➤ *Minor comments:*

Please highlight edited text, rather than simply showing track changes, to indicate revisions. Track changes can be visually confusing, as when deleted figures still appear in the manuscript.

Agreed! We have now changed the way we indicate manuscript revisions to instead highlight the changes we have made.

- *p.3: “The neural representation of transitive...” this does not appear to be a well formed sentence.*
- *p.3-4 And the next sentence an agreement error: “These representations” not “These representation” and “have” not “has”.*

Both sentences have now been fixed:

Studies of the neural representation of transitive and relational reasoning have frequently tested the prediction that relational information is presented in a spatial way, either through the creation of a mental map or a mental model. The representations of mental models of spatial relational reasoning have typically been investigated in hippocampal and entorhinal cortex (Frank et al., 2003 in rats; Heckers et al., 2004; Van Opstal et al., 2008; Van Opstal et al., 2009; Zalesak & Heckers, 2009; Constantinescu et al., 2016; Garvert et al., 2017; Schafer & Schiller, 2017; Theves et al., 2019).

- *p.4. Delete “in rats” or reference the animal model somewhere other than the APA-style in text list of references.*

We have deleted “in rats” because it was ultimately not very important to the overall point.

- *p.4 “2009” should be in brackets for Wendelken et a. 2009.*

Added brackets to 2009:

Specifically, in Wendelken et al. (2009), the authors noted [...]

- *p.4 “In this study, we are primarily...” awkward sentence. Why not just say: “we are primarily interested in testing the prediction of mental model theory that models are separately created and queried”, rather than saying “specifically” twice in the same sentence.*

We agree that the original sentence was overly wordy. We have used your suggested sentence in its place:

In this study, we are primarily interested in testing the prediction of mental model theory that models are separately created and queried (Johnson-Laird, 2010).

- *p.4 “To test this prediction, we separate” when referring to the present study in other places, the authors use past tense. In which case here it should be “we separated”. More generally, the authors should make sure they are being consistent in their use of tense between the old and new text.*

We have updated the tense of our introduction to broadly use the past tense, though we use the present perfect while describing our current study. We also continue to use present tense during declarations of fact rather than discussing prior studies specifically.

- *p.6-7 It seems these details about RSA would fit better in the results. Here it probably suffices to just mention something general about correlating dissimilarity structures.*

We’ve moved the majority of that paragraph into the results section, and have left this portion in the introduction:

In addition to the confounding of spatial content and structure, past research on transitive reasoning tends to use tasks in which participants are able to directly perceive spatial stimuli at the same time as reasoning about it. Whereas this approach is useful for studying the neural basis of the reasoning process, it is sub-optimal for identifying neural patterns that represent inferred mental models unconfounded by perceptual information. Instead, neural patterns representing inferred mental models can be better examined using representational similarity analysis (RSA) to allow researchers to directly measure the representation of the informational content of the reasoning problem, even when the participant is no longer viewing stimuli demonstrating the transitive relationship (as used by Alfred et al., 2018). This approach allows for better understanding of the representation of information at specific points during the reasoning process and allows for a separation of neural processes associated with encoding item-level information versus drawing transitive inferences between items.

- *p.11 Maybe better to say “Participants completed the Hierarchy Recall task in both of two half-hour training sessions”.*

We have adjusted the phrasing of that sentence to match your suggestion:

Participants completed the Hierarchy Recall task in both of two half-hour training sessions and the fMRI session for each content type

- *p.15 here the track changes make things very confusing as it appears as though the same figure occurs twice.*

We have changed from using track changes to highlighting changes. Rest assured, the figure appears only once!

- *p.16-17 the similar amount of representational similarity the authors observe makes quite a lot of sense give the spatial structure was the same across tasks. This is a place where the results of the debriefing are quite instructive. So the authors should*

acknowledge this by explicitly referring to the “common spatial structure of the mental model constructed...”.

We have added a note reminding the readers about the consistent use of a spatial strategy by participants, and how that might affect the pattern of neural results:

In short- the right IPS seems to be representing information about the common structure of the mental model constructed through transitive reasoning rather than the content of the specific transitive reasoning problems. It is not surprising that there is a great deal of overlap in the neural localization of mental models across content types. All the problems used in this study were transitive reasoning problems, which have a very specific organizational structure. Further, participants overwhelmingly reported using a spatial strategy to organize the information in the transitive reasoning problems, such as organizing the information on a line. Given nearly all participants used a spatial strategy for the transitive reasoning problems in all content types, we would predict to see similar broad patterns of activity for all transitive reasoning problems unless the content itself was the primary determiner of patterns of neural activity.

- *p.18-19 I am not sure you need to say much more than you averaged the maps here. Some of the details mentioned here are redundant on those in the methods where they are more carefully described. By this I mean details like converting all other values to 0, why averaging was chosen over conjunction, and that the max possible value is 4.1. I realize I asked previously to foreshadow more of the details in the results, but this level of granularity does not seem necessary.*

Some additional detail from the Method section have been included in the Results section as well, simply due to the fact that the Results section appears before the Method section and we wanted to ensure that readers understood analysis choices that we made. Regardless, we have shortened some of these details (specifically, the maximum possible value and the motivation for averaging over conjunction, especially as we also include conjunction now):

In order to examine the effect of rank, we identified regions that were present in all three of the permutation-corrected content RSA maps. Areas present across all three RSA z-maps are likely regions that represent rank order differences in all possible problem content types (as opposed to being content-specific). To determine candidate regions, we averaged the permutation-corrected z-maps for all three content types, thresholded $z < 1.65$ (one-tailed $p < .05$) to eliminate any values that could have been artificially boosted by chance through the averaging. From the distribution of possible correlations calculated during our permutation correction, we identified that the level of noise in the data to be at a maximum of $z = +/- 1.37$, indicating that a threshold at this level or greater would be sufficient to exclude spurious activation.

- *p.19 Should there be a space between “135mm” and “2 or greater”?*

This error was the result of some superscript formatting being lost. It has been fixed to properly read as mm^2 .

- *p.20 Should be “Instead, we found” not “we find” if tense is to remain consistent with the older text.*

Updated the tense in that sentence:

Whereas a specific location in the rostralateral prefrontal cortex (RLPFC) is frequently reported in studies of transitive reasoning (Bunge et al., 2009; Waechter et al., 2013; Wendelken et al., 2008), we did not find that same region of the RLPFC to be significantly correlated with the average RSA for rank across content types. Instead, we found a cluster that is anterior and ventral to the typically reported region.

- *p.20 Clearly the mental models for the different tasks are equivalent in spatial structure, given what subjects said during debriefing.*

We have added another reminded to readers that this outcome is consistent with the strategies participants reported using:

Since the set of content types range from fully spatial (height) to not inherently spatial but spatializeable (price) to novel (vilchy), regions supporting the mental model of rank order for all these content types did not necessarily need to be spatial. It is important to note, once again, that despite the fact that the content varied from spatial to non-spatial, nearly all participants reported using spatial strategies to organize the structure of the information from these problems.

- *p.23 Why is the definition of the p value changing in the notes for table 1? .05 is very different from .001. Because of track changes, I cannot tell what is going on in this table anymore.*

We have now explained this in more detail in response to the reviewer’s earlier point above, but the primary reason is because in calculating the average RSA map, we previously used the permutation-corrected z-values but did not threshold the inputs to the average RSA map, leading to the desire to use stricter corrections at that point. However, because we now threshold the RSA inputs to the average map as well as spatially cluster-correcting the final map, we reduced the levels of the cluster threshold to account for correcting and thresholding multiple times. In short, all reported clusters are significant at $p < .05$, corrected, and the z-values for the peak individual nodes within a cluster are also compared to a permuted null distribution of correlation values and then thresholded at $z > 1.65$ (which is above our calculated noise threshold). Moreover, this means that the reported z-values in Table 1 and Figure 4 are averages which include zeros for any individual content map that has a sub-threshold value at that node (e.g., all nodes in the aPFC cluster that are active in the conjunction map for *height* and *abstract* include a 0 value for *price*). The table as it exists in the manuscript is pasted here for reference, and we now use highlights instead of track changes to indicate what has been updated, so as to avoid further confusion.

Table 1

Peak Coordinates (MNI) and Anatomical Regions for Average Rank RSA

Anatomical Region	X	Y	Z	Peak Z	Mean Z	Content Overlap
-------------------	---	---	---	--------	--------	-----------------

Intraparietal Sulcus (R)	40	-28	38	2.02	1.39	Height, Price, Abstract
Inferior Frontal Cortex (L)	-40	22	28	2.06	1.43	Height, Price, Abstract
Anterior Prefrontal Cortex (L)	-8	64	0	1.93	1.49	Height, Abstract

Note: Individual content RSA maps were thresholded at the node level based on our noise threshold calculations prior to averaging ($z > 1.65$). Cluster spatial extent thresholds were bootstrapped using AFNI's SurfClust function. Clusters have a minimum area of 120 mm², have a minimum distance of 3mm between nodes in a cluster, and are significant, $p < .05$, corrected. The z -values reported here and in Figure 4 are averages from all 3 content RSA maps which include zeros for any individual content map that has a sub-threshold value at that node.

- *p.27 “identified and tested this link and found...” not “by finding”.*

Fixed tense issue:

Parkinson and colleagues (2014) identified and tested this link and found that literal spatial distance [...]

- *p.28 “information contained...” I think here you can explicitly refer to spatial structure. After doing such a nice job making the spatial content/structure distinction, one might as well use it here.*

Updated to shorten that sentence and use “spatial structure”:

Because mental models tend to be represented spatially, including as mental maps of a given problem space, the spatial structure of these models seems to frequently be represented in neural regions that encode literal spatial maps or spatial distance, such as the hippocampal place maps (Frank et al., 2003; Heckers et al., 2004; Van Opstal et al., 2008; Van Opstal et al., 2009; Zalesak & Heckers, 2009; Constantinescu et al., 2016; Garvert et al., 2017; Schafer & Schiller, 2017; Theves et al., 2019) and the right parietal cortex, especially the right IPS (Krawczyk, 2012; Waechter et al., 2013; Vendetti & Bunge, 2014; Parkinson et al, 2014; Wendelken 2015; Schafer & Schiller, 2017; Alfred et al., 2018).

- *p.28 Missing comma between “normal vision” and “participated in this study”.*

Added the missing comma back in:

Nineteen participants (15 female, $M_{\text{age}} = 19.81$) undergraduate and graduate students, who were right-handed, fluent in English, and with normal or corrected to normal vision, participated in this study.

- *p.33 Figure 6 legend: I am having difficult parsing the first sentence. Should the beginning read “Within each content-type block of each training session, participants...”.*

Fixed the beginning of the sentence to read more clearly:

Within each content type block in each training session, participants completed a set of tasks in a fixed order to learn the hierarchy and practice the tasks they would need to do in the fMRI session.

- *p.33 Figure 6 legend: “In the fMRI session, participants” this is a very awkward sentence. What does “learning review” mean? That subjects just reviewed the material somehow?*

Yes, this referred to just reviewing the information they initially learned in the “Learning” blocks of the training session. We have clarified that in the figure caption for Figure 6:

In the fMRI session, participants completed a short review of the content presented in the Learning portion of the training sessions while anatomical scans were running at the beginning.

- *p.38 “By using averages, we would allow data...” I am not sure you need this sentence to justify averaging.*

We have removed that sentence, especially because we now use both averaging and conjunction analyses to identify regions used for mental models across content types.

- *p.39 was the bootstrap corrected threshold also one-tailed?*

The bootstrapped threshold was two-tailed. To recap the details above, all p -values were two-tailed with the sole exception of the first-level RSA inputs to the 3-way average and conjunction analyses, which were thresholded at the node level above the calculated noise threshold, and which had a further correction applied at the cluster level after averaging across all 3 content RSA maps: a bootstrapped cluster extent threshold of $p < .05$, corrected (two-tailed).

- *p.39 “After the content-specific...” this sentence is difficult to parse. There are some embedded clauses that are hard to recognize as such Please revise. The same is true for “After clusters were identified, we further verified...” There are three embeddings in “we further verified that the clusters we found that we had also hypothesized we would find...”. These sentences could be made much clearer/simpler.*

We have revised those sentences (among others noted earlier) to make them easier to understand:

The content-specific permutation-corrected z -maps that were thresholded at $z > 1.65$ were averaged together to create one average map that represented relative rank of items, across content types. We then used AFNI’s surface-based cluster simulation to identify significant clusters on the surface (clusters would be required to have an area greater than 120 mm^2 , at a bootstrap-corrected threshold of $p < .05$, corrected).

After clusters were identified, we further verified that the clusters were not due to an exceptionally high value from a single content specific map.

Reviewer #3 (Remarks to the Author):

- *This is a revised version of a manuscript I reviewed a while ago. This manuscript aims to investigate the neural encoding of the mental models required for transitive reasoning across different content types from visuospatial to abstract. In this version, the authors have addressed most of my concerns to satisfaction. In particular, the descriptions of the RSA pipeline and relevant details on statistics have been much improved. The additional discussions on existing studies of mental models, on how this study is positioned among these studies, and on what contributions it aims to achieve are also helpful.*

I have one remaining concern on the authors' approach to identify brain regions that encode a similar spatial structure for mental models across content types. I had doubts about using an average z-value across the three content types and suggested using a conjunction instead. In this round of revision, the authors did not embrace this suggestion and offered some explanations justifying averaging, which I do not find convincing. To my understanding, the logic behind the question is quite straightforward – the authors should be looking for a logical “AND” between the clusters encoding each of the content types. Doing an averaging of the effect sizes (z-values) in the individual content types simply does not fulfill this, with or without the additional twists (e.g. thresholding and corrections) that were introduced in the current version. For example, a comparison of Figure 3 and 4 could easily find that there is no aPFC clusters for the “price” content type in Figure 3, while in Figure 4 aPFC is part of the “common brain regions representing mental models across content types” in the authors' own words. Putting the technical details aside, how could one region that is not even encoding the mental model in one content type (based on whatever statistical criteria the authors chose to use) suddenly becomes one that represents mental models across all content types? I still believe that conjunction is the right approach that fits the research question in this study, and that the authors need to grapple with this issue in a more meaningful way given the centrality of this question in their study.

We would like to sincerely thank Reviewer 3 for this comment. In response to this point, we have now included a conjunction analysis, which clearly indicates which regions show 3-way overlap between the content areas, as well as 2-way overlap between 2 of the 3 content areas. This analysis serves to clarify our results, and also is now more consistent with the results of the individual content analyses, much as the reviewer noted. In particular the reviewer is correct that the most anterior PFC cluster is only present for 2 out of the 3 content areas, which is now clearly indicated in Figure 4 and Table 1. We would also like to note that while conducting the conjunction analysis as part of this current revision, we uncovered a problem with our previous

method of cluster correction for the average RSA analysis that was mis-estimating the area that nodes occupy on the cortical surface (which in part contributed to the disconnect between Figs. 3 and 4 that the reviewer noted). This error has been corrected, and the updated figure along with the conjunction results are included in the updated version of the manuscript. We have reported on the outcome in the Results section (pasted below). Again, we thank the reviewer for prompting this analysis and bringing these issues to our attention.

To summarize the key results of the cluster analysis, the cluster in the right IPS is both a statistically significant cluster in the RSA map that averages across content types, and the same cluster also shows overlap between all three content types. The same is true for a cluster in the left inferior frontal cortex (BA 9/44). The majority of each of these clusters is made up of two-way overlaps between content types, but all three content types overlap in the center of the cluster (much like a Venn diagram). We do not necessarily predict that only the same individual surface nodes in the right IPS (or the left IFC) are encoding this structure for all three content types, so it's not problematic that parts of the region are only encoding two of the three content types. Nonetheless, the areas of overlap between all three content types (shown in black in Figure 4) within a focal region indicate the neural similarities in processing all of these transitive reasoning problems. The only other region to emerge in both the conjunction analysis and the average RSA analysis is the aPFC region that shows overlap between the *height* and *abstract* conditions, but not *price*. No other regions survived thresholding for either the conjunction analysis or the average RSA analysis. Finally, in order to use a color key that would make sense for our conjunction map, we have recolored the price condition in all figures in which it appears (Figs. 1-3, 5, and 6) from green to yellow, so that orange, green, and purple could unambiguously be used to indicate two-way overlap between individual conditions, and black is now used for three-way overlap:

Finally, to determine which regions in the average RSA analysis represented similar mental models in each of the three content types, we calculated a conjunction map. This conjunction map highlights regions where either two or three content types had significant values after the initial permutation-based thresholding step. As with the average RSA, following a node-level thresholding step ($z > 1.65$), we applied a bootstrapped spatial cluster correction to the conjunction map to only preserve clusters significant at $p < .05$, corrected. Only the clusters in the right IPS and the left IFC were both significantly large and contained overlap between all three content types. The extent of the three-way overlap in right IPS is limited, however, the area surrounding that overlap is made up of different two-content overlap sections, indicating that adjacent and partially-overlapping regions of this region are coding these spatial mental models. Similarly, the cluster in the left IFC shows sparse three-way overlap, but it is surrounded by Height-Price and Height-Abstract overlapping regions, indicating the left IFC encodes the structure of the mental model across all three content types in the same region, if not precisely the same voxels within that region. The right IPS is the only region that overlaps with the NeuroSynth "spatial" meta-analytic map, consistent with the hypothesis that this region is involved in encoding the spatial structure of mental models across all three content types. The only other region to emerge in both the conjunction analysis and the average RSA analysis is the aPFC region that shows overlap between the Height and Abstract conditions, but not Price. No other regions survived thresholding for either the conjunction analysis or the average RSA analysis.

Figure 4. Common brain regions representing mental model structure across content types. The z -values in this figure are averages of z -values from the content-specific maps, replacing below-threshold values with zeros. The average z -map was then spatially cluster corrected using a bootstrapped cluster extent threshold of $p < .05$, corrected (minimum area = 120 mm^2 ; see Table 1). This average map is superimposed onto a term-based automated meta-analysis generated using NeuroSynth (“spatial” association map) shown in blue, indicating brain regions that are specifically associated with previous fMRI studies of spatial cognition. Inset figures display the conjunction map for each of the significant clusters. Clusters in both the right IPS and left IFC show overlap between all three content types. See note in Table 1 for explanation of average z -value calculation.

Table 1

Peak Coordinates (MNI) and Anatomical Regions for Average Rank RSA

Anatomical Region	X	Y	Z	Peak Z	Mean Z	Content Overlap
Intraparietal Sulcus (R)	40	-28	38	2.02	1.39	Height, Price, Abstract
Inferior Frontal Cortex (L)	-40	22	28	2.06	1.43	Height, Price, Abstract
Anterior Prefrontal Cortex (L)	-8	64	0	1.93	1.49	Height, Abstract

Note: Individual content RSA maps were thresholded at the node level based on our noise threshold calculations prior to averaging ($z > 1.65$). Cluster spatial extent thresholds were bootstrapped using AFNI’s SurfClust function. Clusters have a minimum area of 120 mm^2 , have a minimum distance of 3mm between nodes in a cluster, and are significant, $p < .05$, corrected. The z -values reported here and in Figure 4 are averages from all 3 content RSA maps which include zeros for any individual content map that has a sub-threshold value at that node.

REVIEWERS' COMMENTS:

Reviewer #3 (Remarks to the Author):

This is a manuscript undergoing a 2nd-round revision, and I reviewed its earlier versions a little while ago. In this version, the authors included a new conjunction analysis of the representation of mental models across different content types. This provides better clarifications to their claims on the category-general spatial encoding of mental models, and addresses my remaining concern for the previous version. I have no more comments and am happy to recommend the publication of this manuscript.

Reviewer #3 (Remarks to the Author):

- *This is a manuscript undergoing a 2nd-round revision, and I reviewed its earlier versions a little while ago. In this version, the authors included a new conjunction analysis of the representation of mental models across different content types. This provides better clarifications to their claims on the category-general spatial encoding of mental models, and addresses my remaining concern for the previous version. I have no more comments and am happy to recommend the publication of this manuscript.*

The authors would like to sincerely thank Reviewer 3 for their suggestion to add the conjunction analysis. We believe that it adds clarity to our results and helps convey which regions contain patterns of activity that reflect the structure of mental models across all of the content types.